



# Microphysical, macrophysical and radiative responses of subtropical marine clouds to aerosol injections.

Je-Yun Chun[1], Robert Wood[1], Peter Blossey[1], and Sarah J. Doherty[1,2]

[1]Department of Atmospheric Sciences, University of Washington, Seattle, USA
[2]Cooperative Institute for Climate, Ocean and Ecosystem Studies, University of Washington, Seattle, USA

**Correspondence:** Robert Wood (robwood2@uw.edu)

**Abstract.**

Ship tracks in subtropical marine low clouds are simulated and investigated using large eddy simulations. Five variants of a shallow subtropical stratocumulus-topped marine boundary layer (MBL) are chosen to span a range of background aerosol concentrations and variations in free-tropospheric (FT) moisture. Idealized time-invariant meteorological forcings and approx-
imately steady-state aerosol concentrations constitute the background conditions. We investigate processes controlling cloud microphysical, macrophysical and radiative responses to aerosol injections. For the analysis, we use novel methods to decompose the liquid water path (LWP) adjustment into changes in cloud properties, and the cloud radiative effect (CRE) into contributions by cloud macro- and microphysics. The key results are that (a) the cloud top entrainment rate increases in all cases, with stronger increases for thicker than thinner clouds; (b) the drying and warming induced by increased entrainment
is offset to differing degrees by corresponding responses in surface fluxes, precipitation and radiation; (c) MBL turbulence responds to changes caused by the aerosol perturbation, and this significantly affects cloud macrophysics; (d) across two days' simulation, clouds were brightened in all cases. In a pristine MBL, significant drizzle suppression by aerosol injections results not only in greater water retention, but also in turbulence intensification, leading to a significant increase in cloud amount. In this case, Twomey brightening is strongly augmented by an increase in cloud thickness and cover. In addition, a reduction in the
loss of aerosol through coalescence scavenging more than offsets the entrainment dilution. This interplay precludes estimation of the lifetime of the aerosol perturbation. The combined responses of cloud macro- and microphysics lead to 10-100 times more effective cloud brightening in these cases relative to those in the non-precipitating MBL cases. In moderate and polluted MBLs entrainment enhancement makes the boundary layer drier, warmer and more stratified, leading to a decrease in cloud thickness. Counterintuitively, this LWP response offsets the greatest fraction of the Twomey brightening in a moderately moist
free troposphere. This finding differs from previous studies which found larger offsets in a drier free troposphere, and results from a greater entrainment enhancement in initially thicker clouds, so the offsetting effects are weaker. The injected aerosol lifetime in cases with polluted MBLs is estimated as 2-3 days, which is longer than the estimates from satellite images.



# 1 Introduction

Stratocumulus clouds cover extensive areas of the ocean surface and influence the climate system primarily by enhancing the
reflection of incoming solar radiation back to space (e.g., Hartmann and Short, 1980). That is, clouds are brighter than open
ocean when seen from space. Cloud brightness, i.e., solar reflectivity, is determined by cloud macrophysical properties (cover-
age and thickness), and microphysical properties (droplet size). Anthropogenic pollution increases cloud condensation nuclei
(CCN) concentrations, which leads to more numerous and smaller cloud droplets (Twomey, 1974). For a fixed liquid water path
(LWP), cloud optical thickness increases sublinearly with cloud droplet number concentration ($N_c$), a behavior known as the
Twomey effect (Twomey, 1977). Observational and modeling studies provide convincing evidence that anthropogenic aerosol
forcing by aerosol-cloud interactions masks a significant fraction of the forcing from increases in well mixed greenhouse gases
(e.g., Zelinka et al., 2014; Bellouin et al., 2020; Forster et al., 2021).

Although the Twomey effect results in cloud brightening, cloud macrophysical responses to aerosols (known as cloud *ad-
justments*) remain highly uncertain, and have the potential to enhance or offset Twomey brightening. Cloud fraction may
increase or decrease with aerosol depending on meteorological conditions and the size and concentration of both background
and injected aerosol. In a precipitating boundary layer, aerosol increases reduce cloud droplet size and collision-coalescence
efficiency, leading to precipitation suppression (Wood, 2012). Taken alone, precipitation suppression should allow the retention
of liquid water in clouds, potentially increasing LWP and cloud cover (Albrecht, 1989). In a non-precipitating boundary layer,
on the other hand, water retention is weak. Increased droplet surface area reduces both the timescale for the evaporation of
liquid water (Wang et al., 2003) and the rate of sedimentation of condensate away from cloud top (Bretherton et al., 2007;
Ackerman et al., 2009). Both effects are expected to enhance cloud top entrainment rate (Ackerman et al., 2004). In most cir-
cumstances, entrainment warming and drying results in thinner clouds with lower LWP (Wood, 2007). Cloud adjustments can
therefore be either positive or negative depending upon the nature of the cloud into which aerosol is introduced, the moisture of
the free-tropospheric air overlying the cloud, and the background and added aerosol properties (Ackerman et al., 2004; Wood,
2007; Glassmeier et al., 2021; Hoffmann and Feingold, 2021).

Recent observational and modeling studies of so-called "natural experiments" such as of shipping and pollution plumes
yield a wide range of estimates of the contribution of cloud adjustments to overall aerosol forcing. Observations over polluted
and adjacent unperturbed regions have shown that the average LWP adjustment is negative, ranging from 3% to 20% (Toll
et al., 2019; Diamond et al., 2020; Trofimov et al., 2020), and it may offset up to 30% of the Twomey effect. Other studies
investigated the $N_c$-LWP relationship more generally using satellite observations (Gryspeerdt et al., 2019) and LES modeling
(Glassmeier et al., 2021), estimating that negative LWP adjustments may offset as much as 60% of the Twomey brightening.
Glassmeier et al. (2021), however, argued that estimates using ship track data overestimate albedo increases associated with
by anthropogenically-enhanced $N_c$ by up to 200%, because the the lifetime of the ship track (typically ∼8 hours) is usually
shorter than the timescale by which clouds relax to an equilibrium state (∼20 hours, see Eastman et al. (2016)).

Better constraining the cloud responses to aerosols is of interest because the radiative forcing from only a small change
in the coverage and thickness of stratocumulus clouds is comparable to the warming resulting from doubling atmospheric





carbon dioxide (Randall et al., 1984). This fact, and clouds' known strong sensitivity to aerosol, led to the idea that deliberately injecting CCN into subtropical low marine clouds might enhance cloud albedo and offset global warming (Latham, 1990). This climate intervention approach is commonly referred to as marine cloud brightening (MCB). Modeling studies with GCMs to

test the potential efficacy of MCB have been conducted by enhancing $N_c$ or reducing the effective radius of cloud drops ($r_e$). They have shown that cooling sufficient to offset a significant fraction of global warming caused by doubling of pre-industrial $CO_2$ is potentially achievable (e.g., Latham et al., 2008, 2012; Rasch et al., 2009; Ahlm et al., 2017; Stjern et al., 2018). Salter et al. (2008) estimated that an injection rate of $1.45 \times 10^6$ particles $m^{-2}s^{-1}$ would produce a sufficient Twomey effect to exert a cloud radiative forcing of -3.7 $W\,m^{-2}$, which is comparable to the positive radiative forcing from a doubling of pre-industrial

$CO_2$. GCMs, however, are not able to fully represent the complexity of aerosol-cloud interactions. For instance, most GCMs show that global-scale aerosol perturbation induces positive LWP adjustment on average (Lohmann and Feichter, 2005), which runs counter to observational evidence.

     Large-eddy simulation (LES) models can resolve most processes relevant to aerosol-cloud-precipitation interactions, allowing more accurate simulations of cloud responses to the addition of aerosols, and for attribution of the drivers behind these

adjustments under a range of local meteorological conditions (e.g., Wang and Feingold, 2009; Wang et al., 2011; Berner et al., 2015; Possner et al., 2018). Using LES models, Wang and Feingold (2009) and Wang et al. (2011) showed that sharp gradients in precipitation generated by spatially variable aerosol concentrations induce a mesoscale circulation, which affects cloud properties. Wang et al. (2011) showed that the albedo perturbation produced by aerosol injection strongly depends on the background cloud droplet concentration and meteorological conditions. Using an LES, Berner et al. (2015) successfully simu-

lated an observed ship track in the collapsed marine boundary layer sampled by aircraft. Sensitivity tests with LES models to changes in background aerosol number concentration are consistent with observations: positive LWP adjustments tend to occur in pristine conditions, and negative adjustments in polluted boundary layers (e.g., Lohmann and Feichter, 2005; Wang et al., 2011; Berner et al., 2015). Possner et al. (2018) investigated ship tracks in deep open-cell stratocumuli, showing that although ship tracks are rarely visible in satellite retrievals in deep boundary layers (e.g., Durkee et al., 2000), their radiative effect could

be significant. While these studies have provided useful new insights, there are still many processes in how clouds respond to the addition of CCN in a plume that need to be better quantified; these include the details of entrainment enhancement, the role of other processes controlling cloud properties (e.g., turbulence, surface flux, precipitation, radiation), and the full temporal evolution of cloud responses.

     This study investigates the processes controlling cloud microphysical, macrophysical and radiative responses to aerosol

injection in MBLs under different background aerosol concentrations and a range in lower free troposphere moisture using LES modeling. Two novel methods are used to quantitatively decompose the cloud adjustments into contributions by different processes, and the cloud radiative effect (CRE) into contributions by changes in cloud droplet number concentration ($N_c$), LWP, and cloud fraction (CF).





## 2 Methodology

### 2.1 Model formulation

This study uses the System for Atmospheric Modeling (SAM) version 6.10, a non-hydrostatic anelastic model (Khairoutdinov and Randall, 2003). Three dimensional simulations are used, with a horizontal grid resolution of 50 m and a vertical grid resolution that gradually varies from 5 m at 400-800 m altitude, where clouds form, to 15 m near the surface and to 70 m at the top (1.55 km) of the vertical domain. The model time step is adaptive with a typical value of ~0.5 seconds. Periodic boundary conditions in both the x and y dimensions are used. The upper part of the domain includes a sponge layer to damp gravity waves and prevent artificial reflection from the upper boundary. The subgrid-scale turbulence is represented using a 1.5-order turbulent closure model with a prognostic formulation of turbulent kinetic energy. Advection of all scalars is calculated using an advection scheme that preserves monotonicity (Blossey and Durran, 2008). Total water mixing ratio ($q_t$) and liquid static energy, $S_l = c_p T + gz - Lq_l$, are conserved under phase changes. Here $c_p$ is the specific heat of air at constant pressure, $T$ is the temperature, $z$ is the altitude, $g$ is the gravitational acceleration, $L$ is the latent heat of evaporation of water, and $q_l$ is the liquid water content. Radiation, entrainment mixing, surface fluxes and precipitation influence the prognostic variables. Radiative transfer (shortwave and thermal infrared) is calculated using the Rapid Radiative Transfer Model for GCM applications (RRTMG, Mlawer et al., 1997), which utilizes a droplet effective radius $r_e$ diagnosed from microphysical variables related to cloud. Sensible and latent heat fluxes from the ocean surface are calculated in each gridbox based on Monin-Obukhov Theory.

The two-moment Morrison microphysics scheme (Morrison and Grabowski, 2008) with autoconversion and accretion parameterized using Khairoutdinov and Kogan (2000) predicts the number concentrations and mixing ratios of cloud and rain droplets. When the water vapor mixing ratio $q_v$ is greater than the saturation mixing ratio, condensation is calculated using saturation adjustment. Evaporation of drizzle is explicitly represented, but vapor deposition onto drizzle is not. Ice phase hydrometeor species are not required because the simulation domain is everywhere above the freezing temperature.

A bulk aerosol scheme (Berner et al., 2013) that predicts the number and dry mass of a single lognormal accumulation mode is combined with the modeled cloud microphysics to simulate the aerosol lifecycle and therefore more faithfully represent aerosol-cloud-precipitation interactions. The scheme represents activation (Abdul-Razzak and Ghan, 2000), autoconversion, accretion, precipitation evaporation, scavenging of interstitial aerosol by cloud and rain (see appendix in Berner et al., 2013), droplet sedimentation and surface fluxes. The number and mass fluxes of sea-salt spray is diagnosed based on the wind speed (Clarke et al., 2006) with a single, lognormal accumulation mode with a mean dry diameter of 255 nm. Note that the size of the aerosol produced by surface flux has been corrected from that in (Berner et al., 2013). Cloud droplets are activated from this single lognormal aerosol size distribution, defined by the number and mass of aerosol and the geometric standard deviation of 1.5. Above-inversion aerosol has a prescribed size of 200 nm. There is no representation of the Aitken mode.



## 2.2 Simulation descriptions

Model simulations in this study are based on an idealized stratocumulus case study used in the Cloud Feedbacks Model Intercomparison Project/Global Atmospheric Systems Studies Intercomparison of Large-Eddy and Single-Column Models (CGILS). Among the cases CGILS considered, we focus on S12 (Blossey et al., 2013), which is derived from monthly mean (July 2003) data from the ECMWF Interim Reanalysis (ERA) near the California coast (35 °N, 125 °W). Unlike in Blossey et al. (2013), solar insolation varies diurnally in these simulations, with corresponding changes in solar zenith angle. Although these simulations use constant-in-time Eulerian forcings (Zhang et al., 2012), they are intended to represent the evolution of an air mass after aerosol injection for approximately 40 hours downstream following the aerosol injection. While such an air mass would be expected to experience changing forcings over that time period, we choose steady forcings to characterize the effect of aerosol injection over time on an important MBL cloud regime. In future work, we plan to evaluate the effect of aerosol injection on Lagrangian case studies of stratocumulus to cumulus transitions (e.g., Sandu and Stevens, 2011; Blossey et al., 2021).

This study investigates the effects of aerosol injections into five variants of the CGILS S12 case. The variants are designed to explore how cloud responses depend upon different aerosol background conditions and lower free-troposphere water vapor concentrations. The cases are designed to have aerosol concentrations in the control case (no aerosol injection) that are in approximately steady-state over the two day period. This avoids problems of interpretation that would be inevitable if the background aerosol concentrations are strongly evolving with time, and facilitates attribution of cloud responses to specific processes. An attempt was also made to ensure that boundary layer depth and cloud properties do not strongly drift over time. To avoid strong drifts, lower free-troposphere aerosol $N_{a,FT}$ and large-scale divergence $D$ values were adjusted to produce quasi steady-state conditions in the MBL.

Initial and boundary conditions for the five variant cases are shown in Table 1. All other meteorological drivers are the same across all of the cases. The cases Pristine6, Middle6 and Polluted6 produce different precipitation regimes: precipitating, weakly precipitating, and non-precipitating stratocumulus, respectively. Polluted6, Polluted 3.5, and Polluted1.5 have the same aerosol initial and boundary conditions but have free tropospheric water vapor mixing ratios $q_{FT}$ of 6.0, 3.5 and 1.5 g kg$^{-1}$ respectively. All three of the polluted cases are non-precipitating. Simulations are performed for a 96 km × 9.6 km domain for the Pristine6 and Middle6 cases and for a 48 km × 9.6 km domain for the Polluted6, 3.5 and 1.5 cases. The wider domains for the Pristine6 and Middle6 are used to ensure that a significant portion of the domain remains free of injected aerosol for the entire run to allow an analysis of mesoscale interactions between the track and surrounding clouds, which are more significant in the precipitating cases. All the cases are run for 39.5 hours after spin up, which cover the first daytime and nighttime periods and the second daytime period, which will be referred as to Day 1, Night and Day 2, respectively.

The model is spun up for 12 hours, from 1600-0400 LT. Then, two branched runs with (*Plume*) and without (*Ctrl*) aerosol injection are produced for each case. In the *Plume* runs, a point sprayer travels one time across the center of the x axis (long dimension) along the y axis (short dimension) with domain-relative speed of 10.5 m s$^{-1}$ for 914 seconds. This is consistent with the model domain representing a 9.6 km wide slab of an air mass moving at 10.5 m s$^{-1}$ over a stationary point source injection



site. The mean modal dry diameter of the injected aerosol is 205 nm. The injection rate is $10^{16}$ s$^{-1}$ in Pristine6, $3 \times 10^{16}$ s$^{-1}$

in Middle6 and $3.25 \times 10^{16}$ s$^{-1}$ in the polluted cases in order to produce a roughly consistent fractional increase in the mean aerosol concentration, $\langle N_a \rangle$, across the cases. The injection rates used in this study are comparable to those suggested in previous studies (e.g., Salter et al., 2008; Wang et al., 2011; Wood, 2021). The initial $\langle N_a \rangle$ perturbation ($d \langle N_a \rangle_{\mathrm{init}}$) is given in Table 1.

The quasi steady-state conditions are also summarized in Table 1. The buoyancy jump across the inversion ($\Delta b_t$) is strongest

in the polluted cases where the large-scale subsidence is strongest. The jump of total water mixing ratio ($\Delta q_t$) largely varies with $q_{FT}$, from $-\Delta q_t$ of 3.6-3.8 g kg$^{-1}$ in the moist case to 7.3 g kg$^{-1}$ in the driest case, Polluted1.5. In-cloud droplet concentration ($NCCLD$) is similar to $\langle N_a \rangle$ in the *Ctrl* cases (Table 1).

## 3  Results

### 3.1  General description

In the following two subsections, general characteristics of the cloud fields in the baseline (*Ctrl*) runs and the perturbations to the clouds in the aerosol injection (*Plume*) runs are described.

#### 3.1.1  Baseline (*Ctrl*) runs

Table 2 shows averages and standard deviations of meteorological conditions during Day 1, Night and Day 2 in the *Ctrl* runs for each case. As observed in subtropical marine stratocumulus (Wood et al., 2002), the LES simulations show a strong

diurnal cycle in cloud properties. Figures 1 and 2 show the cloud LWP across the model domain in each case during the Day 1 and Night, respectively, for the *Ctrl* runs. In all cases, mesoscale roll convection develops, similar to that seen in LES simulations of a case study using aircraft-observed fields Berner et al. (2015). In the Pristine6 runs, the roll structure is less coherent, especially during daytime, and the cloud structure more closely resembles open cellular convection. This change in the cloud field organization is likely caused by cloud-base precipitation. Solar absorption may also help to break the roll

structure (Chlond, 1992; Müller and Chlond, 1996; Glendening, 1996; Berner et al., 2015).

During daytime, in-cloud LWP (LWPCLD) is lower than at night due to solar absorption weakening the net radiative cooling at cloud top. This reduces the primary source of turbulence in marine low clouds (Table 2). Notable is that LWPCLD in the Polluted3.5 and Polluted1.5 runs is much lower than 60 g m$^{-2}$, contradicting the argument by Hoffmann et al. (2020) that LWP $< 60$ g m$^{-2}$ is difficult to sustain in steady state. Our inclusion of the diurnal cycle (in contrast to Hoffmann et al. (2020)) may

permit sustaining lower LWP values in our cases, but this is unclear. CF in Pristine6 also varies diurnally from $\sim 50\%$ percent in the day to $\sim 80\%$ percent at night. During the daytime, weak turbulence is unable to supply water vapor from the subcloud to the cloud layer so that the cloud is depleted by drizzle. During the night the turbulence is intensified and cloud water is recovered. The Middle6 and Polluted6 cases remain overcast through the entire day, without stabilization of the boundary layer





due to drizzle evaporation. As the FT becomes drier, CF decreases because the lifting condensation level (LCL) becomes closer
to the inversion due to incorporation of dry FT air into the boundary layer.

For the Pristine6 run, the cloud base precipitation rate $R_{cb}$ varies diurnally from $0.40\,\mathrm{mm\,day^{-1}}$ during the day to $0.81\,\mathrm{mm\,day^{-1}}$
at night. $R_{cb}$ in the Middle6 run is much weaker due to the low coalescence efficiency of the smaller cloud droplets, and this
precipitation is too weak to reach the surface. $R_{cb}$ in the three polluted cases is negligible ($<0.01\,\mathrm{mm\,day^{-1}}$). Similarly, the
surface precipitation rate $R_{sfc}$ in the Pristine6 run changes from $0.15\,\mathrm{mm\,day^{-1}}$ during the daytime to $0.31\,\mathrm{mm\,day^{-1}}$ at
night. $r_e$ is largest in Pristine6, and becomes smaller as $\langle N_a \rangle$ increases. Among the three polluted cases, the cloud droplet
effective radius $r_e$ is largest for the case with the moist FT (Polluted6) due to thicker clouds and a comparable number of cloud
droplets.

Figure 3 shows vertical profiles of $q_t$ and $s_l$ during day and nighttime. Generally, $q_t$ and $s_l$ profiles are more stratified during
daytime than nighttime, because solar absorption at cloud top suppress turbulence by cloud-top cooling. Evaporation of rain
drops below cloud base stabilizes the MBL, so the Pristine6 case is the most stratified of the cases. The MBL is more stratified
with a moister FT than a drier FT, because a drier FT is more transparent to outgoing longwave radiation so that cloud-top
cooling is more effective.

The entrainment rate $w_e$ also varies depending on meteorological conditions (Table 2 and Fig. 4). Factors controlling $w_e$
include boundary-layer turbulence, the strength of the buoyancy jump across the inversion $\Delta b$, the boundary layer depth $z_{inv}$,
and an efficiency term $A$ that incorporates the microphysical (e.g., evaporation, sedimentation) impacts on entrainment. We
use the parametric formula suggested in Bretherton et al. (2007):

$$w_e = \frac{A w_*^3}{z_{inv} \Delta b} \tag{1}$$

where $w_*$ is the convective velocity scale, a measure of the buoyant production of turbulence, defined as the vertical integral of
the buoyancy flux over the boundary layer: $w_* = \left(2.5 \int_0^{z_{inv}} \overline{w'b'} dz\right)^{1/3}$. Here, we define $\overline{B}$, as the vertical integral of buoyancy
production normalized by the boundary layer depth: $\overline{B} = w_*^3/z_{inv}$. Therefore, Eq.(1) becomes:

$$w_e = \frac{A\overline{B}}{\Delta b} \tag{2}$$

Entrainment efficiency ($A$) increases with entrainment-zone cloud liquid water amount, which is largely determined by sed-
imentation velocity and cloud thickness (Bretherton et al., 2007), and with the reduction in buoyancy due to evaporation by
dilution of cloud water with above-inversion air (e.g., Nicholls and Turton, 1986). The expressions in equations (1) and (2) are
approximate. If, for example, entrainment is related to a different metric of boundary layer turbulence than $\overline{B}$ (e.g., Stevens,
2002), the computation of $A$ from simulated values of $w_e$, $\overline{B}$ and $\Delta b$ will be affected. As a result, changes in $A$ with aerosol
perturbations may result from both changes in entrainment efficiency and errors in the approximations embedded in equa-
tion (1). Figure 4 shows run-average $A$, $\overline{B}$, $\Delta b$, cloud liquid water mixing ratio at $z_{inv}$-50 m ($q_{c,inv}$) and $w_e$ in the *Ctrl* runs.
Generally, $A$ is proportional to $q_{c,inv}$, consistent with the results of Bretherton et al. (2007).

In the Pristine6 *Ctrl* run $w_e$ is significantly weaker than the other runs, mainly due to small $A$ and $\overline{B}$. The low $A$ may
be attributed to low $q_{c,inv}$ caused by the high sedimentation velocity of large cloud droplets, while the low $\overline{B}$ is caused by




suppression of turbulence from rain formation warming the cloud layer and precipitation evaporation cooling below cloud. Across the weakly- and non-precipitating cases, $w_e$ is quite similar, but the factors driving $w_e$ are different. In the Middle6 and Polluted6 runs, where the FT is moist, $A$ is high but $\overline{B}$ is low. The high $A$ is mainly due to larger $q_{c,inv}$, while the moist

overlying FT leads to weaker cloud top radiative cooling and low $\overline{B}$ (e.g., Siems et al., 1990). For drier FTs, $A$ and $q_{c,inv}$ decrease as clouds become thinner, but $\overline{B}$ increases as cloud-top cooling becomes more effective. For all cases, $w_e$ is greater at night than during the day, due to stronger net radiative cooling at cloud top. In the Pristine6 run, $w_e$ at night ($2.72 \ \mathrm{mm\,s}^{-1}$) is three times greater than during the day ($0.92 \ \mathrm{mm\,s}^{-1}$). This is partly attributed to the large variation in CF, in addition to the stronger net radiative cooling at night.

**3.1.2    Aerosol injection (*Plume*) runs**

Figure 5 shows Hovmöller plots of $\langle N_a \rangle$ (upper row) and LWP (lower row) averaged along the y axis for the *Plume* runs for each case. For all the cases, the ship tracks are approximately parallel to the roll axis, which strongly affects the lateral dilution of the plume (Berner et al., 2015). Except in the Pristine6 run, the plume edge tends to align with the mesoscale cell boundaries, showing that the mixing across the width of a cell is rapid, while that to adjacent cells is relatively slow.

During Day1 in the Pristine6 run, there is a fringe of clear sky at the edges of the plume (Fig. 5f), consistent with Wang and Feingold (2009) and Wang et al. (2011). These cloud-cleared regions are caused by a mesoscale circulation, generated by the gradient of precipitation rates between the region with higher $NCCLD$, due to the injected plume, and the background. The widths of the cloud-cleared regions become broader up until the early afternoon, and then they narrow. At night, the roll structure develops within the plume and the cloud-free fringes disappear. Clouds within the plume become overcast, while

those in the background remain patchy but thick. On Day 2, the areas within the plume remain mostly overcast, while in the background regions the clouds are more broken. Starting at sunrise on Day 2, $\langle N_a \rangle$ in the background air starts to be depleted, likely because turbulence after sunrise is too weak to sustain cloud number concentration against the loss by coalescence. In the other cases, there is no visible change in cloud morphology with aerosol injection.

**3.2    Responses to aerosol injection**

The impacts of aerosol injections are analyzed below using budget equations to quantitatively compare the roles of different processes in the *Plume* and *Ctrl* runs. Throughout, we use $d$ in front of each variable to indicate the *Plume - Ctrl* difference. Comparisons are made for averages over Day 1, Night and Day 2, respectively (Table 3). Values in square brackets indicate the standard deviation from the time series of the domain-mean differences, and provides a measure of the robustness of the *Plume - Ctrl* differences.

**3.2.1    Microphysical impacts on entrainment and turbulence**

In all the cases, domain mean cloud droplet effective radius ($r_e$) robustly decreases with aerosol injection (Table 3), with stronger decreases when unperturbed $r_e$ is large. The decrease in $r_e$ is greatest in the Pristine6 case (-1.3, -1.7 and -1.6 $\mu m$





in Day 1, Night and Day 2, respectively), followed by the Middle6 case (-0.9, -0.9 and -1.6 $\mu m$) in Day 1, Night and Day 2, respectively), despite lower aerosol injection rates in these runs. Among the three polluted cases, the $r_e$ decrease is larger in the moister cases, since unperturbed $r_e$ is larger; the increase in $N_c$ is very similar in all cases (Table 1).

In the Pristine6 and Middle6 cases, the cloud-base precipitation rate $R_{cb}$ decreases in response to aerosol injection. Reduction in droplet size (Table 3) reduces collision-coalescence efficiency. $dR_{cb}$ in the Pristine6 case is -0.06, -0.26, and -0.04 $\mathrm{mm\,day^{-1}}$ on Day 1, Night and Day 2, respectively. $dR_{cb}$ in the Middle6 case is -0.03, -0.07 and -0.06 $\mathrm{mm\,day^{-1}}$. The decrease is larger during nighttime and early morning because the background precipitation rate is higher. Since there is negligible precipitation in the three polluted cases, $dR_{cb}$ is also negligible (Table 3). Likewise, $R_{sfc}$ also decreases in the Pristine6 case (-0.02, -0.09, and -0.01 $\mathrm{mm\,day^{-1}}$ on Day 1, Night and Day 2, respectively). Precipitation is too weak to reach the surface in the Middle6 case, so $dR_{sfc} \sim 0$.

As a result of the reduction in $r_e$, the cloud top entrainment rate $w_e$ is enhanced in all cases due to the sedimentation-entrainment feedback (Table 3). Turbulence production through drizzle suppression (Wood, 2007) is more effective than the sedimentation-entrainment feedback in enhancing turbulence (Bretherton et al., 2007), so $dw_e$ is greatest in the Pristine6 case (Table 3), where $w_e$ in the *Plume* run is increased by 30-100 percent over the *Ctrl* run. The increase in $w_e$ in the Middle6 case is likely also aided by $R_{cb}$ suppression. For the three polluted cases, $dw_e$ is weaker than in the two precipitating cases, but it generally increases with FT moisture.

From Eq.(2), the relative change in entrainment rate can be expressed as:

$$\frac{dw_e}{w_e} = \frac{dA}{A} + \frac{d\overline{B}}{\overline{B}} - \frac{d\Delta b}{\Delta b}. \tag{3}$$

Run-averaged values of $dw_e/w_e$, $dA/A$, $d\Delta b/\Delta b$ and $d\overline{B}/\overline{B}$ (Fig. 6) clearly show that entrainment changes are largely driven by changes in entrainment efficiency $dA$. These changes are much greater in the Pristine6 and Middle6 cases than in the three polluted cases (note the different axes), probably due to the greater reduction in $r_e$ and thus also sedimentation velocity. Likewise, among the three polluted cases, $dA/A$ is greater for a moister than a drier FT, which is also attributed to a greater reduction in $r_e$. Reduced turbulence (i.e., negative $d\overline{B}/\overline{B}$) partly mitigates the $A$-driven increase of $dw_e$. This is largely a damping of the MBL turbulence by the increased entrainment (Stevens, 2002), although in the Pristine6 case, $d\overline{B} > 0$ due to turbulence invigoration from precipitation suppression. $d\overline{B}/\overline{B}$ is more negative in the Polluted6 case than in the Polluted3.5 case, likely because $dA/A$ is greater, and the associated enhancement of entrainment results in increased stratification in the boundary layer.

In contrast to the other polluted cases, $d\overline{B}/\overline{B}$ in the Polluted1.5 case is positive. This is mainly attributed to greater cloud cover in the *Plume* run than in the *Ctrl* run (Table 3), leading to a stronger longwave radiative cooling and thus an intensification of turbulence. An increase in cloud thickness with greater entrainment can happen in a mixed layer (Randall, 1984) but typically requires a moist FT or deeper MBL. Here, increased $\overline{B}$ is occurring with an extremely dry FT, which one would expect to be detrimental to cloud amount. This behavior may be due to a response (increase) in surface fluxes, as discussed in the following section. Finally, it should be noted that $d\Delta b/\Delta b$ contributes only weakly to $dw_e$. In all cases, $d\Delta b$ increases because stronger entrainment tends to sharpen the inversion.





These results imply that, in general, entrainment enhancement in weakly- and non-precipitating MBLs is mainly caused by an increase in entrainment efficiency, which leads to stronger stratification of the boundary layer. In a precipitating MBL, although there is also a large increase in entrainment efficiency, buoyancy production by suppression of precipitation leads to a

more turbulent boundary layer. The combination of the enhancement in $A$ and $\overline{B}$ drive a large enhancement in the entrainment rate.

### 3.2.2    Mean and Coupling states of $q_t$ and $s_l$

As illustrated in Appendix A, LWP adjustments are controlled by changes in $z_{inv}$ and $z_{cb}$, which are determined by both the mean and coupling state of $q_t$ and $s_l$. In this subsection, we show how the mean and coupling states of these variables respond to

aerosol injections and what processes control the responses. To understand the different response of $\langle q_t \rangle$ and $\langle s_l \rangle$, perturbations in net fluxes of $q_t$ and $s_l$ into the boundary layer, averaged throughout the simulation ($dF_{q_t}$ and $dF_{s_l}$, respectively), are given in Fig. 7. Bars represent $dF_{q_t}$ and $dF_{s_l}$ by entrainment (orange), by the rest of the processes (blue), and the sum of all terms (hatched). The tables below show the net fluxes due to different processes (i.e., entrainment (ENT), surface flux (SFX), sedimentation (SED) and radiation (RAD), and sum of all terms (SUM)). The methods to calculate the net fluxes are given in

Appendix A.

The change in fluxes from entrainment, $dF_{q_t,ENT}$ and $dF_{s_l,ENT}$, in the Pristine6 case are -5.9 and 5.9 $\mathrm{W\,m^2}$, respectively, which are about 5-6 times greater than in the the Middle6 case and about one order of magnitude greater than in the Polluted cases. Among the three Polluted cases, $dF_{q_t,ENT}$ is greatest for Polluted1.5, because $\Delta q_t$ is more negative for the drier FT. Since $\Delta s_l$ is comparable across all these cases (Table 1), $dF_{s_l,ENT}$ is mostly explained by $dw_e$.

All the other processes, such as the changes in the surface flux, sedimentation and radiation, buffer the drying and warming of the boundary layer by entrainment enhancement. This buffering is most effective in the Pristine6 case, with surface moisture fluxes $dF_{q_t,SFX}$ and precipitation suppression $dF_{q_t,SED}$ together offsetting almost two thirds of the entrainment drying. Similarly, $dF_{s_l,SFX}$, $dF_{s_l,SED}$, and $dF_{s_l,RAD}$ together offset 84% of the entrainment warming. The surface flux responses and the negative $dF_{s_l,SFX}$ response are attributable both to changes in the MBL mean state and to changes in the degree

of MBL coupling. The suppression of precipitation also induces moistening and cooling of the boundary layer (i.e., positive $dF_{q_t,SED}$ and negative $dF_{s_l,SED}$, respectively). For the Pristine6 case, greater MBL cooling through the radiative flux response (i.e., negative $dF_{s_l,RAD}$) is caused by an enhanced CF at night in the *Plume* run.

The buffering is less effective in the Middle6 case, where only 20% of the entrainment drying is offset by a reduced precipitation flux, and the contribution from surface evaporation is negligible. Approximately 60% of entrainment warming is

buffered by a reduction in surface sensible heat flux, reduced precipitation warming, and increased radiative cooling. Decoupling of the MBL likely plays a role. Although the MBL becomes drier, decoupling suppresses heat and momentum transport, inducing only weak damping from the surface fluxes. Moistening and cooling by precipitation suppression is less effective in the Middle6 case than in the Pristine6 case, due to weaker suppression of $R_{sfc}$. Since $dCF$ is negligible in the Middle6 case, the increased radiative flux divergence in this case is driven by reduced solar absorption from a reduced cloud LWP (Table 3).





The amount of buffering of entrainment drying by other responses depends on the FT moisture. Of the Polluted cases, $dF_{q_t,ENT}$ is most negative for Polluted1.5 due to the drier FT. However, this drying is largely offset by $dF_{q_t,SFX}$ so that $dF_{q_t,SUM}$ is comparable to that in the Polluted3.5 case but weaker than that in the moister Polluted6 case. This can also be explained by the response in the coupling state of the MBL. The greater enhancement of entrainment in the moister FT induces greater stratification, making the damping by surface fluxes less effective. $dF_{s_l,SFX}$, on the other hand, is more sensitive to

$\langle s_l \rangle$, rather than to decoupling. $dF_{s_l,RAD}$ is more negative in the Polluted6 case than in the Polluted3.5 case, probably because of the greater decrease in LWPCLD (e.g., Table 3) leads to a greater reduction in solar absorption. $dF_{s_l,RAD}$ in the Polluted1.5 case is, however, much greater than in the Polluted6 and Polluted3.5 cases. This occurs because radiative cooling strengthens with greater nighttime CF in the *Plume* than the *Ctrl* run (Table 3).

These results imply that drying and warming of the MBL by entrainment is controlled not only by $\Delta q_t$ and $\Delta s_l$, but also by

the response of the entrainment rate to aerosol perturbation. In addition, the surface fluxes, precipitation and radiation respond to aerosol-induced changes in the MBL state and play important roles in the MBL $q_t$ and $s_l$ budgets. This suggests that studies of aerosol cloud-interactions of a day or longer should include interactive surface fluxes so that the buffering mechanisms seen here would be represented.

### 3.2.3   Cloud liquid water path adjustment

The previous sections showed how the mean value and MBL coupling state of $q_t$ and $s_l$ respond to aerosol injection, and what processes contribute to the response. In this section, we will analyze how changes in cloud and MBL properties affect cloud LWP adjustments. Here, we assume that changes in the domain-mean cloud liquid water are determined by changes in cloud fraction ($dLWP_{CF}$), cloud thickness ($dLWP_h$) and cloud adiabaticity ($LWP_{f_{ad}}$). $dLWP_h$ is further decomposed into contributions to $dLWP$ through changes in cloud thickness resulting from a response in the MBL mean state ($dLWP_{MEAN}$)

and in the degree of MBL coupling ($dLWP_{CPL}$). Based on this, cloud LWP adjustments can be decomposed as follows:

$$dLWP = dLWP_{MEAN} + dLWP_{CPL} + dLWP_{CF} + dLWP_{f_{ad}} \tag{4}$$

Details of how each term is calculated are given in Appendix A. Bars in Fig. 8 show $dLWP$ caused by changes in the cloud thickness from responses in the MBL mean state and the coupling state, in CF, and in adiabaticity, as well as the sum of all the terms during (a) Day 1, (b) Night and (c) Day 2.

On Day 1, $dLWP_{MEAN}$ is negative in all cases and is most negative for the Pristine6 case, followed by the Middle6, Polluted3.5 and Polluted1.5 cases. This is consistent with the degree of entrainment enhancement across the cases, indicating that MBL warming and drying by aerosol injection are mainly driven by entrainment enhancement. At Night, $dLWP_{MEAN}$ becomes more negative, except in the Polluted1.5 case. This implies that the systems continues to moves towards a drier and warmer steady state, which is consistent with the results of the Glassmeier et al. (2021) simulations, which showed that the

adjustment equilibrium time scale of cloud macrophysics (about one day) is much longer than that of the cloud microphysics (5 to 10 minutes).





On Day 2, $d\text{LWP}_{MEAN}$ in the Pristine6 case becomes positive (4.66 g m$^{-2}$). This sign change in $d\text{LWP}_{MEAN}$ is mainly attributed to greater drizzle suppression at Night. Drizzle suppression is ineffective during daytime, since background $R_{cb}$ is weak (e.g., Table 2). Therefore, entrainment drying and warming is stronger than the the damping effects discussed above. At Night, drizzle starts to be strongly suppressed (e.g., Table 3), leading to significant liquid water retention and turbulence generation. Thus, $d\text{LWP}$ is driven more strongly by changes in sedimentation (i.e., surface precipitation), surface fluxes and radiation during Night. As a result, by the end of the Night period, entrainment drying and warming is more than offset by these responses, leading to more positive $d\text{LWP}$. Accumulated drizzle suppression during nighttime leads to significant positive $d\text{LWP}_{MEAN}$ on Day 2. This leads not only to thicker clouds, but also to greater CF. During Day 1 and Night, $d\text{LWP}_{CF}$ is negligible (0.26 and 0.29 g m$^{-2}$). On Day 2, $d\text{LWP}_{CF}$ becomes 8.44 g m$^{-2}$, and this term dominates $d\text{LWP}$. Without significant suppression of precipitation, the weakly- and non-precipitating cases do not have this sign change in $d\text{LWP}_{MEAN}$.

Changes in the MBL coupling state in response to aerosol injection also have significant impacts on $d\text{LWP}$. Since turbulence in the boundary layer is intensified by drizzle suppression, $d\text{LWP}_{CPL}$ is positive in the Pristine6 case . More precipitation is suppressed at Night, making $d\text{LWP}_{CPL}$ more positive at Night than during daytime. During daytime in the Middle6, Polluted6 and Polluted3.5 cases, $d\text{LWP}_{CPL}$ is negative due to increased stratification of the boundary layer. $d\text{LWP}_{CPL}$ tends to be more negative when the background clouds are thick (Table 2), due to a significant enhancement in the entrainment efficiency (i.e., Fig.4) leading to greater stratification. At Night, however, $d\text{LWP}_{CPL}$ in these cases becomes negligible, as background drizzle is intensified. In the Polluted1.5 case, where the background LWP is very low, $d\text{LWP}_{CPL}$ is negligible during daytime. At night it becomes positive, and this term dominates total cloud LWP changes.

In the Pristine6 and Middle6 cases, where $R_{cb}$ is not negligible, suppressed $R_{cb}$ by aerosol injections reduces the loss of cloud liquid water by drizzle, leading to an increase in adiabaticity of MBL clouds (Wood, 2005). However, Since there is no precipitation in the Polluted cases, there is of course no reduction in $R_{cb}$ (Table 2), and $d\text{LWP}_{f_{ad}}$ is insignificant.

The overall LWP adjustment $d\text{LWP}_{SUM}$ in the Pristine6 case is positive, despite significant enhancement of $w_e$, due to responses in the coupling state of the MBL and increase in CF and $f_{ad}$ offsetting entrainment drying. $d\text{LWP}_{SUM}$ increases with time, as drizzle suppression is accumulated. $d\text{LWP}_{SUM}$ during daytime in the Middle6 case, where background LWP is greatest, is the most negative of all the cases, and become less negative as background LWP is lower. $d\text{LWP}_{SUM}$ is more negative on Day 2 than on Day 1 in the weakly- and non-precipitating MBLs (i.e. all cases other than Pristine6). The Middle6 case stands out in having a relatively smaller negative $d\text{LWP}_{SUM}$ at Night than during the Days, compared to the other cases, This is because the background $R_{cb}$ in this case is larger so that positive values of $d\text{LWP}_{CPL}$ and $d\text{LWP}_{f_{ad}}$ compensate for the negative $d\text{LWP}_{MEAN}$. In the Polluted1.5 case, where the background clouds are quite thin, the LWP adjustment is much weaker.

### 3.2.4 Aerosol and cloud number concentration

The lifetime of aerosol perturbations is an important factor for determining the full extent of cloud radiative responses to injections. To understand the evolution of the aerosol perturbations, Fig. 9 shows the budget terms for entrainment (ENT), autoconversion (AUT), accretion (ACC), scavenging (SCV), sedimentation (SED) and surface flux (SFX), with the values in



Fig.9 corresponding to differences of $\langle N_a \rangle$ between the *Plume* and *Ctrl* runs, divided by the initial perturbation, $(\widehat{d\langle N_a \rangle})$. For example, a unit of -1 $\mathrm{cm}^{-3}/\mathrm{cm}^{-3}$ means the initial perturbation has dissipated completely over the time of the simulation (39.5 hours), while +1 $\mathrm{cm}^{-3}/\mathrm{cm}^{-3}$ means the initial aerosol perturbation has doubled.

Because the aerosol injections increase $\langle N_a \rangle$ considerably, dilution by cleaner FT air acts as a sink and negative $d\langle N_a \rangle$.

$\widehat{d\langle N_a \rangle}_{ENT}$ is smaller in the Pristine6 case than in the other cases (about -0.1 $\mathrm{cm}^{-3}/\mathrm{cm}^{-3}$). This is mainly because the free troposphere is more polluted than the boundary layer, and the initial perturbation is not significant compared to the jump of $N_a$ across the inversion (See Table 1). In the other cases, where the boundary layer is more polluted than the free troposphere, $\widehat{d\langle N_a \rangle}_{ENT}$ act as a major sink, ranging from -0.45 to -0.47 $\mathrm{cm}^{-3}/\mathrm{cm}^{-3}$.

Another primary cause of differences in $\widehat{d\langle N_a \rangle}$ across cases is collision-coalescence. For precipitating clouds, aerosol in-
jections reduce coalescence scavenging, so that the sink of aerosol to accretion and autoconversion decreases. In the Pristine6 case, where precipitation is most strongly suppressed, accretion and autoconversion together induce positive $\widehat{d\langle N_a \rangle}$ (contributing 0.70 and 0.07 $\mathrm{cm}^{-3}/\mathrm{cm}^{-3}$, respectively). This implies that in a strongly precipitating MBL, aerosol injection may induce a transition from open- to closed cells. In a weakly precipitating case (Middle6), $\widehat{d\langle N_a \rangle}$ due to change in accretion and auto-conversion is positive but smaller (0.07 and 0.12 $\mathrm{cm}^{-3}/\mathrm{cm}^{-3}$, respectively), extending the lifetime of the aerosol perturbation.
In non-precipitating cases, these effects are negligible.

Scavenging of aerosol by coagulation onto cloud droplets induces universally negative $\widehat{d\langle N_a \rangle}$, because the higher concentration of $N_a$ produced by aerosol injection leads to a higher scavenging rate. However, these terms do not have a large impact in the overall $\widehat{d\langle N_a \rangle}$ in any of the cases examined. Surface aerosol fluxes and sedimentation of rain and cloud droplets do not respond significantly to aerosol injection, so this does not contribute to $\widehat{d\langle N_a \rangle}$.

Based on the budget approach presented here, we can infer the lifetime of aerosol perturbations. The sum of all the terms (hatched bar in Fig. 9, $\widehat{d\langle N_a \rangle}_{SUM}$) represents the change of $\widehat{d\langle N_a \rangle}$ over the duration of the experiment after injection (39.5 hr). Thus, the lifetime can be roughly estimated by dividing 39.5 hours by $\widehat{d\langle N_a \rangle}_{SUM}$. Since $\widehat{d\langle N_a \rangle}$ increases with time for the Pristine6 case, it is not possible to estimate a lifetime for injected aerosol in this case. The injected aerosol lifetime in the Middle6 case is ~90 hr, and for the Polluted cases is ~65 hr. These timescales are considerably larger than the typical age
(~7 hr) of ship tracks detected using satellite observations (Durkee et al., 2000). Coupled with the fact that the magnitude of the cloud LWP adjustments become stronger on the second day than on the first day (Fig. 8), this points to the need to track cloud responses to aerosol injections over multiple days in order to provide an assessment of their overall radiative effect. After two days, most ship tracks will have lost their identity and it may be difficult to distinguish regions containing injected aerosol from marine background conditions.

**3.2.5 Cloud Radiative Effect**

Previous sections have illustrated how cloud LWP and cloud droplet number concentrations respond to aerosol injections for the different background meteorological and aerosol cases. This section analyzes injection-induced changes in the cloud radiative effect ($d\mathrm{CRE}$) by decomposition into contributions from changes in $N_c$ ($d\mathrm{CRE}_{N_c}$), i.e., the Twomey effect, and from those due to adjustments in cloud LWP ($d\mathrm{CRE}_{LWP}$) and CF ($d\mathrm{CRE}_{CF}$). The decomposition approach is described in Appendix B.




In all the cases, the residuals (RES) between the actual $d$CRE and the sum of all components are much smaller than individual components, indicating that the decomposition works quite well. Values of $d$CRE given here are an average over 24 hours, so it accounts for the zero values during nighttime. When comparing $d$CRE across cases, note that the domain size in the Pristine6 and Middle6 cases is twice that in the Polluted cases, and that the amount of aerosol injection is different among the cases (see Table 1). The bar plot in Fig. 10 summarizes $d$CRE$_{N_c}$, $d$CRE$_{LWP}$ and $d$CRE$_{CF}$ on Day 1 and Day 2 for all of the cases.

For the Pristine6 case, $d$CRE$_{N_c}$, $d$CRE$_{LWP}$ and $d$CRE$_{CF}$ on Day 1 are -5.8 and -0.8 and -3.3 $\mathrm{W\,m^{-2}}$, respectively. Positive adjustments in both LWPCLD and CF (Table 3) approximately double the brightening induced by the Twomey effect alone. On Day 2, contributions from all three cloud properties increase in magnitude (Fig. 10), with $d$CRE$_{CF}$ dominating the brightening.

For all other cases, cloud fraction responses $d$CRE$_{CF}$ contribute only minimally or not at all to $d$CRE, and CRE changes largely comprise Twomey effects and LWP adjustments only. In all cases other than the Pristine6 case, LWP adjustments 425 are negative ($d$CRE$_{LWP} > 0$). In the Middle6 case, LWP reductions offset almost 50% of Twomey brightening on Day 1. On Day 2, there is a small amount of cloud brightening in the morning, but cloud becomes darkened afternoon, which offsets cloud brightening earlier in the morning (Fig.S7b). The actual $d$CRE during Day 2 of Middle6 indicates brightening, even though the sum of the $N_c$, LWP and CF contributions to $d$CRE indicate darkening. This discrepancy is attributed to the near cancellation of the $N_c$ and LWP contributions, which allows errors in predicting the individual contributions to dominate the total. For the 430 three Polluted cases, $d$CRE$_{N_c}$ is similar across cases, and in all cases is slightly smaller on Day 2 than Day 1. Although a more evenly distributed plume would result in greater Twomey brightening on Day 2, this is offset by the fact that the magnitude of the aerosol perturbation is decreasing with time (Fig. 9). The negative LWP adjustments in the polluted cases offset 10-30% of the Twomey effect on Day 1, increasing to 20-50% on Day 2. It is noteworthy that $d$CRE$_{LWP}$ is more positive under a moist FT (Fig. 10), a response that appears to differ from Glassmeier et al. (2021), wherein the strongest negative LWP adjustments 435 occur with a very dry FT. This is due to the coupling and surface flux responses described above in earlier sections.

## 4 Discussion

Our results highlight the complexity of cloud responses to aerosol injections. Changes in the mean state and the coupling state of the MBL, in cloud cover and in the adiabaticity of MBL clouds (Fig. 8) are important for determining the cloud radiative responses. Cloud responses depend on the background conditions, such as background aerosol loading, cloud thickness and 440 cover, and free-tropospheric moisture. A key result is that the aerosol-induced cloud top entrainment rate increases more rapidly for thicker than for thinner clouds (Fig. 11a) because entrainment efficiency, which is an estimate of how strong entrainment is for a given level of turbulence, is more strongly affected by cloud thickness (Hoffmann et al., 2020; Zhang et al., 2021) than by the dryness of the FT. In weakly- and non-precipitating MBLs (i.e., the Middle6, Polluted6, 3.5 and 1.5 cases), entrainment enhancement ($dw_e$) monotonically increases with unperturbed background LWP (Fig. 11a). This result seems to be broadly 445 consistent with Possner et al. (2020) who show that the LWP susceptibility to cloud number concentration is more negative when the boundary layer is deeper and so clouds are thicker. In a strongly-precipitating MBL (e.g. the Pristine6 case here), turbulent intensification by suppression of drizzle evaporation greatly augments the increase in entrainment efficiency. In



response to enhanced entrainment, perturbations in surface fluxes, radiative fluxes at cloud top, and precipitation all offset entrainment-enhanced drying and warming of the MBL. In addition, changes in MBL stratification (quantified here as changes in coupling, cloud cover and adiabaticity) by entrainment enhancement (drizzle suppression) also affect cloud macrophysics and thus, cloud radiative properties.

Figure 11b shows the impact of background LWP on the ratio of the radiative effects of cloud adjustments (LWP plus CF) to the Twomey effect, i.e., RLT=$(d\mathrm{CRE}_{\mathrm{LWP}}+d\mathrm{CRE}_{\mathrm{CF}})/d\mathrm{CRE}_{\mathrm{N_c}}$. For example, RLT=-1 indicates that the Twomey effect is exactly cancelled by cloud adjustments, whereas for RLT=+1 the adjustments produce a doubling of the brightening compared with the Twomey effect alone. For weakly- and non-precipitating MBLs, RLT tends to linearly decrease with background LWP (Fig. 11b). On Day 2, the slope of the line becomes more negative, as the system moves towards an equilibrium steady state (Glassmeier et al., 2021). In the strongly-precipitating regime, the LWP adjustment becomes more positive, so that RLT becomes positive.

One can define a brightening efficiency as the total additional solar energy reflected per injected particle. The per-particle efficiency decreases by over an order of magnitude as background $N_c$ increases from 10 to 100 cm$^{-3}$ (Fig. 11c). This is consistent with results from other LES studies and is somewhat steeper than that from a simple heuristic model with no cloud adjustments (see Fig. 4d in Wood (2021)). Positive LWP adjustment at low $N_c$ and negative adjustments under more polluted conditions steepen this curve compared with expectations from the Twomey effect alone. This nonlinearity means that assessments of the potential global forcing from MCB (e.g., Wood, 2021) should ideally consider the temporal variability of the background cloud droplet concentration and aerosol. Such a strong sensitivity of the brightening to the unperturbed aerosol state also suggests that there may be considerable benefit in targeting injections to occur primarily in regimes with very low aerosol concentrations. These regimes of extreme albedo susceptibility likely occur relatively infrequently, but could provide a significant fraction of the overall MCB radiative forcing. However, rare extreme brightening events likely make it challenging to assess MCB efficacy for a region as a whole.

The lifetime of aerosol perturbations is also examined here, since this will directly impact the duration of cloud radiative effects. In strongly drizzling MBLs, aerosol injection significantly reduces coalescence scavenging losses by cloud and rain drops, enough to surpass the enhanced loss by entrainment dilution (Fig. 9). This leads to increasing aerosol and cloud number concentration in the MBL, at least over the first 2 days, making it impossible to define an injected aerosol lifetime. Even in a weakly drizzling MBL, aerosol lifetime can be extended by slowing the rate at which the aerosol perturbation is damped. Our simulations show that the aerosol lifetime in a non-precipitating MBL, which is mainly determined by entrainment dilution, is about 65 hours, which is much longer than the typical longevity of ship tracks seen in satellite observations (e.g. Durkee et al., 2000; Gryspeerdt et al., 2021). One possible reason is that ship track identification using satellite images is essentially based on the variation in cloud number concentration over the track, which sharply decreases with time mainly due to lateral dilution (plume dispersion) (Berner et al., 2015) rather than loss of injected aerosol by microphysical processes. As the aerosol plume spreads and dilutes, such aerosol perturbations will become more difficult to detect above the noise caused by spatial variability of background clouds. A lack of track detectability does not necessarily mean a lack of radiative forcing, however. More work is required to understand how satellite observations can be used together with LES to assess track detectability and radiative





forcing for those cases where the injected aerosol had spread and diluted markedly. Aerosol lifetime might also depend on the jump in aerosol number concentration across the inversion, which is not investigated in this study. However, given the small increase in the cloud-top entrainment rate with aerosol perturbation in non-precipitating MBLs (i.e. 2-5 percent, shown in Fig. 6), the jump of aerosol number concentration might have a marginal impact on the lifetime.

Figure 12 conceptualizes the findings in this study. It illustrates the responses in precipitation, entrainment rate, boundary layer turbulence and surface fluxes to aerosol injection under different meteorological conditions. In a strongly-drizzling MBL (Fig.12a), drizzle is strongly suppressed by aerosol injection. This induces a large enhancement in the entrainment rate, both by increased entrainment efficiency and by turbulent invigoration. As a result, there is considerable entrainment drying and warming of the MBL. However, these effects are largely countered by other processes. First, turbulent invigoration by suppressed sub-cloud precipitation evaporation together with greater retention of cloud liquid water in the cloud both drive better MBL coupling. In addition, drying and warming of the MBL are largely offset by negative surface sensible and latent heat flux feedbacks, reduced moisture loss from surface precipitation, and increased longwave radiative flux divergence. The combination of these effects results in large increases in cloud thickness, cover and adiabaticity, further enhancing brightening from the Twomey effect.

The responses of clouds in weakly- and non-precipitating MBLs are distinctly different. In these cases, the entrainment rate is enhanced by the cloud droplet sedimentation-entrainment feedback. Because drizzle suppression is weak or negligible, it provides no source of turbulent invigoration. Instead, MBL turbulence tends to weaken in response to aerosol injections because enhanced entrainment reduces the buoyancy flux (Stevens, 2002). This leads to increased MBL decoupling. In addition, weaker turbulence makes surface flux moistening/cooling feedbacks less effective. Therefore, a combination of entrainment drying/warming and increased MBL stratification makes clouds thinner, reducing cloud LWP. Figure 12b and c illustrate the steady-state adjustment of clouds to aerosol injection under moist and dry FT conditions, respectively. Under a moister FT without strong drizzle (Fig. 12b), background clouds are thick. Aerosol injection into thick clouds considerably enhances the entrainment rate. Without the strong damping effect of surface fluxes, the MBL becomes drier and warmer. Furthermore, the MBL is more stratified, leading to weaker moisture and temperature fluxes through cloud base and thus, greater cloud thinning. Under a dry FT (Fig. 12c), on the other hand, background clouds are thin and the sedimentation-evaporation feedback in thin clouds leads to a weaker enhancement of entrainment. This leads to weak drying/warming of the boundary layer and little induced stratification and thus, only a small reduction in cloud thickness.

These results are in contrast with previous observational (e.g., Gryspeerdt et al., 2019; Possner et al., 2020) and modeling studies (e.g., Wood, 2007; Glassmeier et al., 2021), where increasing aerosol concentrations in clouds with a dry overlying lower FT induces large reduction in cloud LWP. This difference might be because our simulations are for a specific regime of stratocumulus clouds – those in a shallow and mostly coupled MBL. LWP increases more rapidly with MBL depth under a dry FT due to more effective cloud-top cooling, so mixing (e.g., Eastman and Wood, 2018; Possner et al., 2020) and negative LWP adjustment under a dry FT might become more negative rapidly with MBL depth compared to that under moist FT. A recent modeling study by Glassmeier et al. (2021) found that in an extremely dry FT the Twomey effect could be entirely cancelled by negative LWP adjusments (cloud thinning), possibly leading to cloud darkening. That study noted that the time-scale of





cloud macrophysics (about 20 hours) is much longer than that of response of cloud microphysics to aerosol perturbation (less than one hour). As such, they conclude that the radiative forcing derived for ship tracks, which are mostly observed only a
few hours after injection, may represent an overestimate the overall cooling effect of the ship track, as it misses the negative LWP adjustments that affect the clouds over several days. However, their simulations used fixed latent and sensible heat fluxes from the ocean surface, which is one of the main processes that offsets the drying and warming of the MBL by enhanced entrainment in this study. Since the time-scale of these damping effects is comparable to that of cloud macrophysics, their estimate of negative LWP adjustment in a steady-state equilibrium may be overestimated.

Future work should investigate other cloud regimes, such as deeper cumulus clouds, using high-resolution, process-resolving models. Our simulations only cover shallow, well-mixed stratocumulus clouds, which is not the most dominant cloud regime over the subtropical and Tropical oceans. Thus, in order to reduce uncertainty of global cloud radiative forcing caused by anthropogenic aerosol perturbation and to evaluate the potential efficacy of MCB, we also need to investigate the processes occurring in different MBL regimes. It is expected that the processes described here may be quite different in other cloud
regimes.

## 5    Conclusions

Our limited understanding of aerosol-cloud interactions accounts for a considerable uncertainty in anthropogenic aerosol radiative forcing. To attempt to overcome challenges associated with entangled aerosol and meteorological influences on clouds (Stevens and Feingold, 2009), many studies have utilized 'natural experiments' such as ship tracks and cloud responses to emis-
sions from volcanic eruptions and power plants (Christensen et al., 2021). These real-world aerosol-induced cloud responses have provided invaluable insights into aerosol-cloud interactions, since we can directly compare the cloud properties with and without aerosol perturbation under the same meteorological conditions. However, these studies show a large variability in responses, implying that cloud responses are strongly dependent on the meteorological conditions, as well as to the background and injected aerosol properties. This motivates the investigation of aerosol-cloud interactions under a variety of meteorological
conditions using LES modeling.

This study investigates the MBL and cloud responses to aerosol injections in LES simulations in a shallow, idealized stratocumulus-topped MBL near the California coast (35 °N, 125 °W). We focus on the effects of different background aerosol loading and levels of free-tropospheric moisture, using two-day simulations of five cases with different background aerosol number concentrations, free-tropospheric moisture and large-scale subsidence. Across two days of simulation, cases
with clean, moderately polluted and polluted MBLs and a moderately moist (6 g kg$^{-1}$) lower FT and a polluted case with a dry (1.5 g kg$^{-1}$) lower FT showed cloud brightening, cloud radiative effects ranging from -3 to -35 W/m2 across the cases and simulation days. A moderately polluted case with higher FT moisture (6.0 g kg$^{-1}$) had cloud brightening on the first day and second day morning, then cloud darkening in the afternoon on the second day, which cancels almost all the brightening in the morning on the second day.



Our results show that aerosol injection affects the cloud-top entrainment rate, MBL turbulence, surface fluxes and cloud microphysics differently depending on meteorological conditions. These responses alter the mean and coupling states of the MBL, which both play important roles in the cloud adjustment.

The followings are the key responses modulating the LWP adjustment:

1) Aerosol injection enhances the cloud-top entrainment rate through the entrainment-sedimentation feedback (Bretherton
et al., 2007). This enhancement is larger when the unperturbed clouds are thick, since then cloud drop size is more susceptible to aerosol perturbation, leading to increased entrainment efficiency. In precipitating conditions, suppression of cloud base precipitation greatly enhances the entrainment rate (Bretherton et al., 2007).

2) Enhanced entrainment induces MBL drying and warming, which is substantially damped by perturbations in surface latent and sensible heat fluxes, by radiative flux at cloud top and by surface precipitation. The damping effect by surface fluxes
is more effective when the boundary layer is well-mixed. When drizzle is greatly suppressed, increased cloud cover (which increases radiative cooling) and increasing cloud adiabaticity both augment the damping.

3) The response of the MBL coupling state to aerosol injection strongly depends on the background meteorological conditions. In a strongly precipitating MBL, the suppression of sub-cloud drizzle evaporation induces stronger coupling of the MBL, leading to an increased cloud amount and LWP. In weakly- and non-precipitating MBLs, aerosol injections cause stronger MBL
stratification by entraining more buoyant air from the free troposphere; this leads to a decreased cloud LWP. The greater entrainment enhancement is, the more stratified the MBL becomes. The stratification effect is shown to be more significant during daytime than nighttime.

4) In a strongly-precipitating MBL, precipitation suppression significantly reduces aerosol loss by coalescence scavenging, leading to differences in aerosol concentrations between the runs with (*Plume*) and the runs with (*Control*) runs that increase
with time over the two day simulation. This implies an infinite effective lifetime for injected aerosol. The effective aerosol lifetime in non-precipitating MBLs is estimated at 65 hours; this is much longer than the estimates of ship track lifetimes made using satellite images (e.g., Durkee et al., 2000; Gryspeerdt et al., 2021). Even a weak suppression of precipitation significantly extends the effective lifetime of aerosol perturbations.

5) In all cases examined, aerosol injections into shallow marine clouds induce Twomey brightening, augmented by the
positive LWP adjustment in a pristine MBL with strong drizzle and offset by negative LWP adjustments in moderate and polluted MBLs. Counterintuitively, Twomey brightening was more strongly offset by negative LWP adjustments when the FT was moister. Clouds in a strongly-precipitating MBL become brighter with time following aerosol injection, while the brightening in weak- and non-precipitating MBLs decrease on the second day of simulation. It is therefore possible that there could be cloud dimming beyond the 2-day duration of our simulations.

Given the multi-day effective aerosol lifetime of injected aerosol, together with the finding that cloud LWP responses (be they positive or negative) grow with time, we may conclude that even two-day simulations are not of a sufficient duration to fully capture marine low cloud responses to point-source aerosol injections. In reality, meteorological boundary conditions will not be constant for such long periods, and so a future simulation strategy is required that allows for quasi-idealized evolution





of boundary conditions (e.g., increasing SST over time). Such simulations will more realistically capture the evolution of
background cloud fields over the subtropical eastern oceans.

**Appendix A: Decomposition of the LWP adjustment**

Mixed-layer theory first proposed by Lilly (1968) has been widely used to model stratocumulus-topped boundary layers and
is a simple but useful tool to investigate microphysical and macrophysical processes controlling cloud thickness (e.g., Randall
et al., 1984; Wood, 2007; Hoffmann et al., 2020). However, the fundamental assumption in the mixed layer model that the
boundary layer is fully mixed and liquid water adiabatically increases with height without consideration of cloud cover is not
an accurate portrayal of reality in many cases. Here, we use a modified model which predicts LWP responses to changes in
cloud thickness and cover, and the adiabaticity of clouds as follows:

$$LWP = CF \int_{z_{cb}}^{z_{inv}} \rho q_l dz = \frac{1}{2} CF \rho f_{ad} \Gamma_{q_l} h^2 \tag{A1}$$

Here, $\Gamma_{q_l}$ is the lapse rate of the liquid water content, $z_{inv}$ is inversion height, $f_{ad}$ is adiabaticity, and $h$ is cloud thickness.
$f_{ad}$ is calculated as an average of adiabaticity at every cloudy column. For the calculation of adiabaticity for individual cloudy
columns, cloud thickness $h$ is defined as a vertical thickness of grids where $N_c$ is greater than 0.1 cm$^{-3}$. Assuming that the
aerosol perturbation changes $h$, CF and $f_{ad}$, the resulting adjustment of LWP can be expressed as follows:

$$
\begin{aligned}
dLWP &= \frac{1}{2}(CF + dCF)\,\rho\,(f_{ad} + df_{ad})\,\Gamma_{q_l}\,(h + dh)^2 - \frac{1}{2}\,CF\,\rho\,f_{ad}\,\Gamma_{q_l}\,h^2 \\
&\approx CF\,\rho\,f_{ad}\,\Gamma_{q_l}\,h\,dh + \frac{1}{2}\,\rho\,(f_{ad} + df_{ad})\,\Gamma_{q_l}\,(h + dh)^2\,dCF + \frac{1}{2}\rho\,CF\,\Gamma_{q_l}\,(h + dh)^2 df_{ad} \\
&\equiv dLWP_h + dLWP_{CF} + dLWP_{f_{ad}}
\end{aligned} \tag{A2}
$$


where $d$ represents the difference between the *Plume* and *Ctrl* runs.

The LWP adjustment due to changes in cloud thickness LWP$_h$ can be further decompoosed because cloud thickness ad-
justments arise from changes in both $z_{inv}$ and $z_{cb}$, i.e., $dh = dz_{inv} - dz_{cb}$. We assume that the cloud base height change $dz_{cb}$
is determined by the moisture $q_t^c$ and liquid static energy $s_l^c$ of the cloud layer, rather than from MBL-mean values of these
variables ($\langle q_t \rangle$ and $\langle s_l \rangle$ respectively). These variables are decomposed into their mean through the depth of the boundary layer
and their residual:

$$q_t^c = \langle q_t \rangle + \delta q_t \tag{A3a}$$

$$s_l^c = \langle s_l \rangle + \delta s_l \tag{A3b}$$

where $\delta$, the residual, is the difference between the cloud layer and the MBL mean, a measure of the MBL decoupling. Cloud
layer values are calclated for the upper third of the cloud layer within cloudy columns. The values of $\langle q_t \rangle$ and $\langle s_l \rangle$ change in
response to fluxes into the boundary layer such as entrainment, surface fluxes, radiation and surface precipitation, while $\delta q_t$
and $\delta s_l$ vary with the coupling state of the boundary layer.





The cloud base height $z_{cb}$ is equal to the LCL defined by $q_t^c$ and $s_l^c$ as described below. The response of $z_{cb}$ can be expressed using the two moist conserved variables as in Wood (2007).

$$
\begin{aligned}
dz_{cb} &= \frac{\partial z_{cb}}{\partial q_t} dq_t^c + \frac{\partial z_{cb}}{\partial s_l} ds_l^c \\
&= \frac{\partial z_{cb}}{\partial q_t} \left( d\langle q_t \rangle + d\delta q_t \right) + \frac{\partial z_{cb}}{\partial s_l} \left( d\langle s_l \rangle + d\delta s_l \right).
\end{aligned}
\tag{A4}
$$

From this, $d\mathrm{LWP}_h$ can be calculated as:

$$
\begin{aligned}
d\mathrm{LWP}_h &= CF\, \rho\, f_{ad}\, \Gamma_{q_l} h(dz_{inv} - dz_{cb}) \\
&= CF\, \rho\, f_{ad}\, \Gamma_{q_l} h \left[ \left( dz_{inv} - \frac{\partial z_{cb}}{\partial q_t} d\langle q_t \rangle - \frac{\partial z_{cb}}{\partial s_l} d\langle s_l \rangle \right) + \left( -\frac{\partial z_{cb}}{\partial q_t} d\delta q_t - \frac{\partial z_{cb}}{\partial s_l} d\delta s_l \right) \right]
\end{aligned}
$$

$$
\equiv d\mathrm{LWP}_{MEAN} + d\mathrm{LWP}_{CPL}
\tag{A5}
$$

The term in the first parentheses on the right-hand side corresponds to the LWP adjustment through a change in the MBL mean state $d\mathrm{LWP}_{MEAN}$, while the second corresponds to the LWP adjustment through a change in the coupling state $d\mathrm{LWP}_{CPL}$.

Combination of Eqs. A2 and A5 indicates that cloud LWP changes are determined by changes in cloud thickness associated with changes in the MBL mean state ($d\mathrm{LWP}_{MEAN}$), coupling state ($d\mathrm{LWP}_{CPL}$), cloud cover ($d\mathrm{LWP}_{CF}$) and adiabaticity

($d\mathrm{LWP}_{f_{ad}}$):

$$
d\mathrm{LWP} = d\mathrm{LWP}_{MEAN} + d\mathrm{LWP}_{CPL} + d\mathrm{LWP}_{CF} + d\mathrm{LWP}_{f_{ad}}
\tag{A6}
$$

The variables needed to calculate these terms are estimated using domain-averaged vertical profiles from the LES. $z_{inv}$ is identified as the level where the product of the vertical gradients in moisture and in temperature is at a minimum. $z_{cb}$ is the lowest height at which $q_l > 0.01\,\mathrm{g\,kg}^{-1}$. The values of $\Gamma_{q_l}$, $\partial z_{cb}/q_t$, and $\partial z_{cb}/\partial s_l$ are estimated as in Wood (2007):

$$
\Gamma_{q_l} = \frac{c_p}{L_v}(\Gamma_d - \Gamma_s)
\tag{A7}
$$

$$
\partial z_{cb}/\partial q_t = -(R_a T_{cb}/g q_t)[(L_v R_a/c_p R_v T_{cb}) - 1]
\tag{A8}
$$

$$
\partial z_{cb}/\partial s_l = 1/g
\tag{A9}
$$

Here, $\Gamma_d$ and $\Gamma_s$ are the dry and moist adiabatic lapse rates, respectively; $c_p$ is the specific heat at constant pressure; $R_a$ and $R_v$ are the specific heats of dry air and water vapor, respectively; $T_{cb}$ is temperature at cloud base; $g$ is Earth's gravitational

acceleration.

## Appendix B: Decomposition of Cloud Radiative Effect

We use a novel method to decompose the change in the cloud radiative effect ($d\mathrm{CRE}$) into the components caused by changes in cloud droplet number concentration ($d\mathrm{CRE}_{N_c}$), LWP ($d\mathrm{CRE}_{\mathrm{LWP}}$) and CF ($d\mathrm{CRE}_{\mathrm{CF}}$). The derivation of the method is based on the equations in Diamond et al. (2020), and the calculations are conducted using LES outputs of solar insolation ($F_\odot$),

cloudy-sky and clear-sky net shortwave radiative flux at the top of atmosphere ($F_{cld}$ and $F_{clr}$), and in-cloud $N_c$ and LWP.





The overall change in the cloud radiative effect by an aerosol perturbation ($d$CRE) can be defined as:

$$d\text{CRE} = CRE_{pl} - CRE_{ctrl} = F_\odot(A_{pl} - A_{ctrl}),$$ (B1)

where $A$ is the domain-mean albedo and the subscripts $pl$ and $ctrl$ denote the runs with and without aerosol perturbation, respectively. The domain contains a mixture of cloudy and clear columns. Therefore, $A$ can be decomposed into contributions from the clear-sky ($A_{clr} = (F_\odot - F_{clr})/F_\odot$) and cloudy-sky ($A_{cld} = (F_\odot - F_{cld})/F_\odot$) regions as follows:

$$A = CF\,A_{cld} + (1 - CF)A_{clr}$$ (B2)

Using Eq. B2, Eq .B1 can be converted as follows:

$$d\text{CRE} = F_\odot(CF_{pl}\,A_{cld,pl} + (1 - CF_{pl})\,A_{clr} - CF_{ctrl}\,(A_{cld,ctrl} - (1 - CF_{ctrl})A_{clr})$$ (B3)

$$= F_\odot(\underbrace{CF_{ctrl}\,(A_{cld,pl} - A_{cld,ctrl})}_{d\text{CRE}_{cld}} + \underbrace{(CF_{pl} - CF_{ctrl})\,(A_{cld,pl} - A_{clr})}_{d\text{CRE}_{CF}}))$$

The first term on the right hand side represents change in the CRE due to change in cloud optical thickness ($d\text{CRE}_{cld}$), while the second represents the effect of changes in cloud fraction ($d\text{CRE}_{CF}$).

The next step is to decompose $d\text{CRE}_{cld}$ further into $d\text{CRE}_{N_c}$ and $d\text{CRE}_{LWP}$. To do this, we first need to estimate changes in cloud albedo ($\alpha$) due to changes in $N_c$ and LWP:

$$d\alpha = d\alpha_{Nc} + d\alpha_{LWP} = dN_c\frac{\partial\alpha}{\partial N_c} + d\text{LWP}\frac{\partial\alpha}{\partial\text{LWP}}$$ (B4)

where $d\alpha_{N_c}$ and $d\alpha_{LWP}$ are cloud albedo changes due to change in $N_c$ and LWP, respectively. Using cloud albedo susceptibility to $N_c$ and LWP (Platnick and Twomey, 1994; Quaas et al., 2008):

$$\frac{\partial\alpha}{\partial N_c} = \frac{1}{3}\frac{\alpha(1-\alpha)}{N_c}$$ (B5a)

$$\frac{\partial\alpha}{\partial\text{LWP}} = \frac{5}{6}\frac{\alpha(1-\alpha)}{\text{LWP}}$$ (B5b)

We can explicitly calculate the perturbation in cloud albedo due to changes in $N_c$ and LWP as follows:

$$\alpha_{N_c,pl} = \int_{N_{c,ctrl}}^{N_{c,pl}}\frac{\partial\alpha}{\partial N_c}dN_c + \alpha_{ctrl} = \frac{\alpha_{ctrl}r_{N_c}'^{1/3}}{\alpha_{ctrl}(r_{N_c}'^{1/3} - 1) + 1}$$ (B6a)

$$\alpha_{LWP,pl} = \int_{\text{LWP}_{ctrl}}^{\text{LWP}_{pl}}\frac{\partial\alpha}{\partial\text{LWP}}d\text{LWP} + \alpha_{ctrl} = \frac{\alpha_{ctrl}r_{LWP}'^{5/6}}{\alpha_{ctrl}(r_{LWP}'^{5/6} - 1) + 1}$$ (B6b)

where $r_N' = N_{c,pl}/N_{c,ctrl}$; $r_{LWP}' = \text{LWP}_{pl}/\text{LWP}_{ctrl}$; $\alpha_{ctrl}$ denotes cloud albedo without perturbation, which can be calculated using a simplied single-layer atmospheric model for solar radiation (Donohoe and Battisti, 2011; Qu and Hall, 2005):

$$\alpha = \frac{A_{cld} - \alpha_{atm}}{T^2 + \alpha_{atm}A_{cld} - \alpha_{atm}^2}$$ (B7)



where $T$ is transmissivity of the atmosphere (i.e. $T = F_{clr}/F_\odot$); $\alpha_{atm}$ is the albedo of the atmosphere. To convert cloud albedo ($\alpha_{Nc,pl}$ and $\alpha_{LWP,pl}$) to overcast albedo ($A_{Nc,pl}$ and $A_{LWP,pl}$), we use a rearranged Eq.B7:

$$A = \alpha_{atm} + \alpha \frac{T^2}{(1 - \alpha_{atm}\alpha)}. \tag{B8}$$

Then, Eq.B3 is further decomposed into

$$dCRE = F_\odot(\underbrace{CF_{ctrl}\left(A_{cld,Nc} - A_{cld,ctrl}\right)}_{dCRE_{N_c}} + \underbrace{CF_{ctrl}\left(A_{cld,LWP} - A_{cld,ctrl}\right)}_{dCRE_{LWP}} + \underbrace{(CF_{pl} - CF_{ctrl})\left(A_{cld,pl} - A_{clr}\right)}_{dCRE_{CF}})$$

However, the aerosol plumes do not cover the whole domain (Fig. 5). Thus, in order to accurately quantify the $dCRE$ between the *Plume* and *ctrl* runs, we need to separate the plume and background as follows:

$$dCRE = F_\odot\big[\underbrace{CF_{ctrl}\left[AF_{pl}(A_{cld,Nc,pl} - A_{cld,ctrl}) + AF_{bg}(A_{cld,Nc,bg} - A_{cld,ctrl})\right]}_{dCRE_{Nc}}$$

$$+ \underbrace{CF_{ctrl}\left[AF_{pl}(A_{cld,LWP,pl} - A_{cld,ctrl}) + AF_{bg}(A_{cld,LWP,bg} - A_{cld,ctrl})\right]}_{dCRE_{LWP}}$$

$$+ \underbrace{AF_{pl}(A_{cld,pl} - A_{cld,ctrl})(CF_{pl} - CF_{ctrl}) + AF_{bg}(A_{cld,bg} - A_{cld,ctrl})(CF_{bg} - CF_{ctrl})}_{dCRE_{CF}}\big) \tag{B9}$$

where $AF_{pl}$ and $AF_{bg}$ represent the areal fractions of the track and background respectively in the *pl* run. Validation of the above method by comparing the calculated $dCRE$ with model output is provided in the supplementary materials.

*Code and data availability.*    All code and data used in this study are available on request from the authors

*Author contributions.*    JY, RW and PB formulated the original model study. JY and PB set up the model runs, and JY conducted and analyzed the runs, with inputs from RW, PB and SD. JY drafted the paper, and RW, PB and SD provided edits and revisions.

*Competing interests.*    The authors have no competing interests.

*Acknowledgements.*    This study was primarily supported by NOAA's Climate Program Office Earth's Radiation Budget (ERB) Program, Grant NA20OAR4320271, as well as with funding from Lowercarbon, the Pritzker Innovation Fund, and SilverLining, through the Marine Cloud Brightening Project. This publication is also partially funded by the Cooperative Institute for Climate, Ocean and Ecosystem Studies (CICOES) under NOAA Cooperative Agreement NA15OAR4320063, Contribution No. 2022-1200.



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



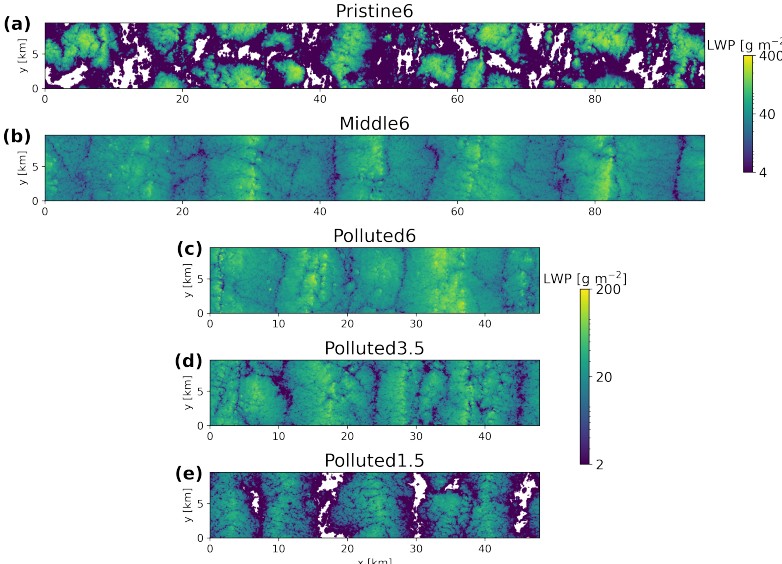

**Figure 1.** Horizontal fields of LWP at local noon (LT 1208) on Day 1 in the *Ctrl* run for the Pristine6, Middle6, Polluted6, Polluted3.5 and Polluted1.5 cases.

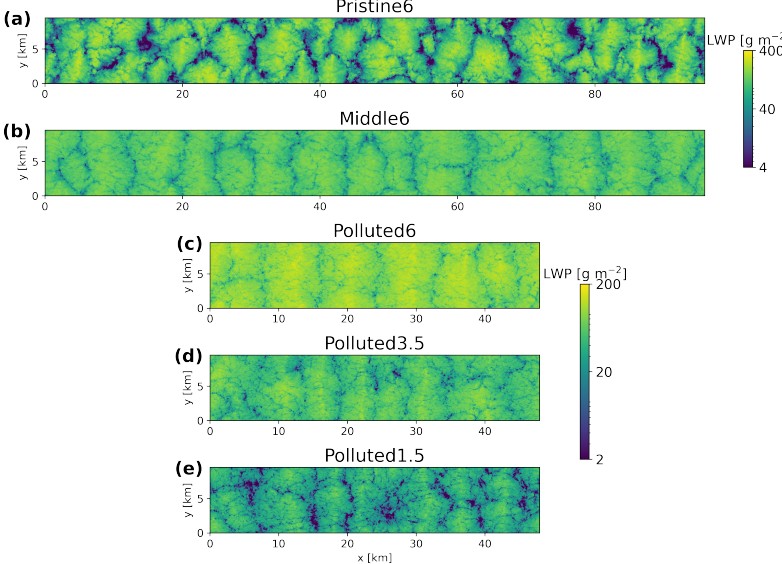

**Figure 2.** As in Fig.1, but for nighttime (LT 0008).

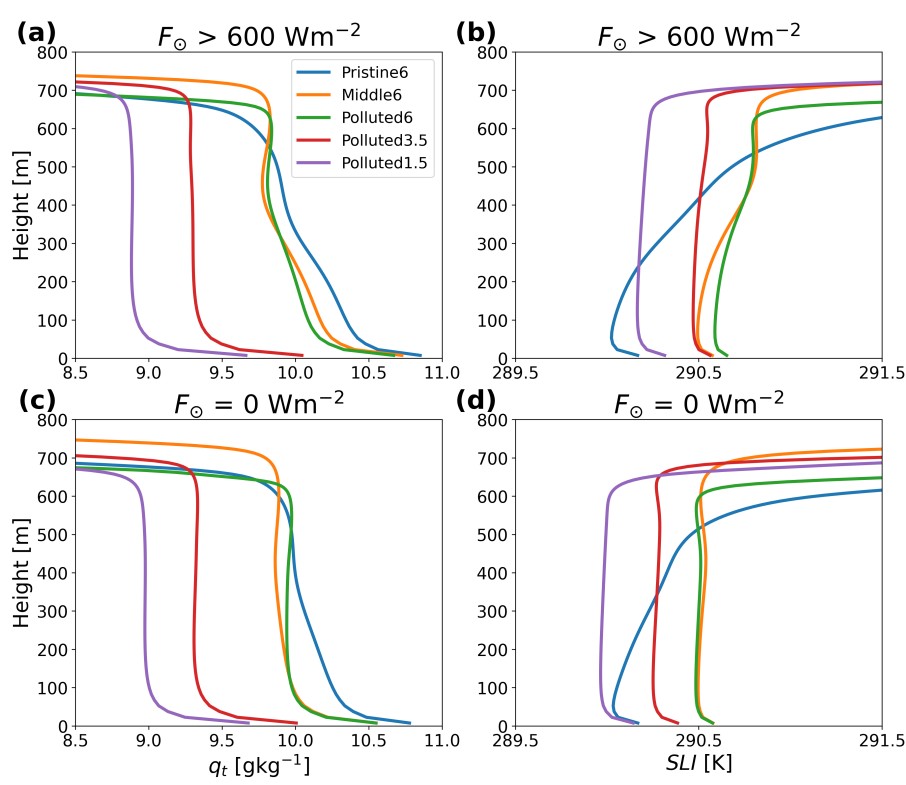

**Figure 3.** Vertical profiles of (left column) $q_t$ and (right column) $s_l$ averaged during (upper row) day and (lower row) nighttime.





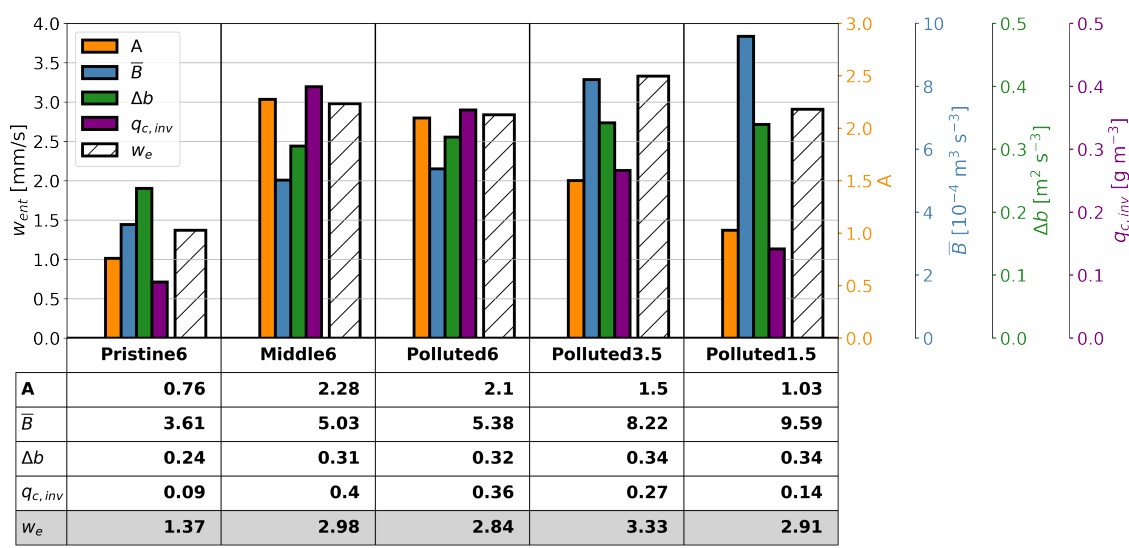

| | Pristine6 | Middle6 | Polluted6 | Polluted3.5 | Polluted1.5 |
|---|---|---|---|---|---|
| A | 0.76 | 2.28 | 2.1 | 1.5 | 1.03 |
| $\overline{B}$ | 3.61 | 5.03 | 5.38 | 8.22 | 9.59 |
| $\Delta b$ | 0.24 | 0.31 | 0.32 | 0.34 | 0.34 |
| $q_{c,inv}$ | 0.09 | 0.4 | 0.36 | 0.27 | 0.14 |
| $w_e$ | 1.37 | 2.98 | 2.84 | 3.33 | 2.91 |

**Figure 4.** Run-average entrainment velocity ($w_e$, hatches), entrainment efficiency (A, orange), boundary-layer-mean buoyancy flux ($\overline{B}$, blue), buoyancy jump ($\Delta b$, dark green) and cloud water mixing ratio at $z_{inv} - 50\text{m}$ ($q_{c,inv}$, purple) in the *Ctrl* runs for the Pristine6, Middle6, Polluted6, Polluted3.5 and Polluted1.5 cases. The y-axis on the left hand side is for $w_e$, and the different colored axes on the right hand side correspond to bars of the same colors in the figure.

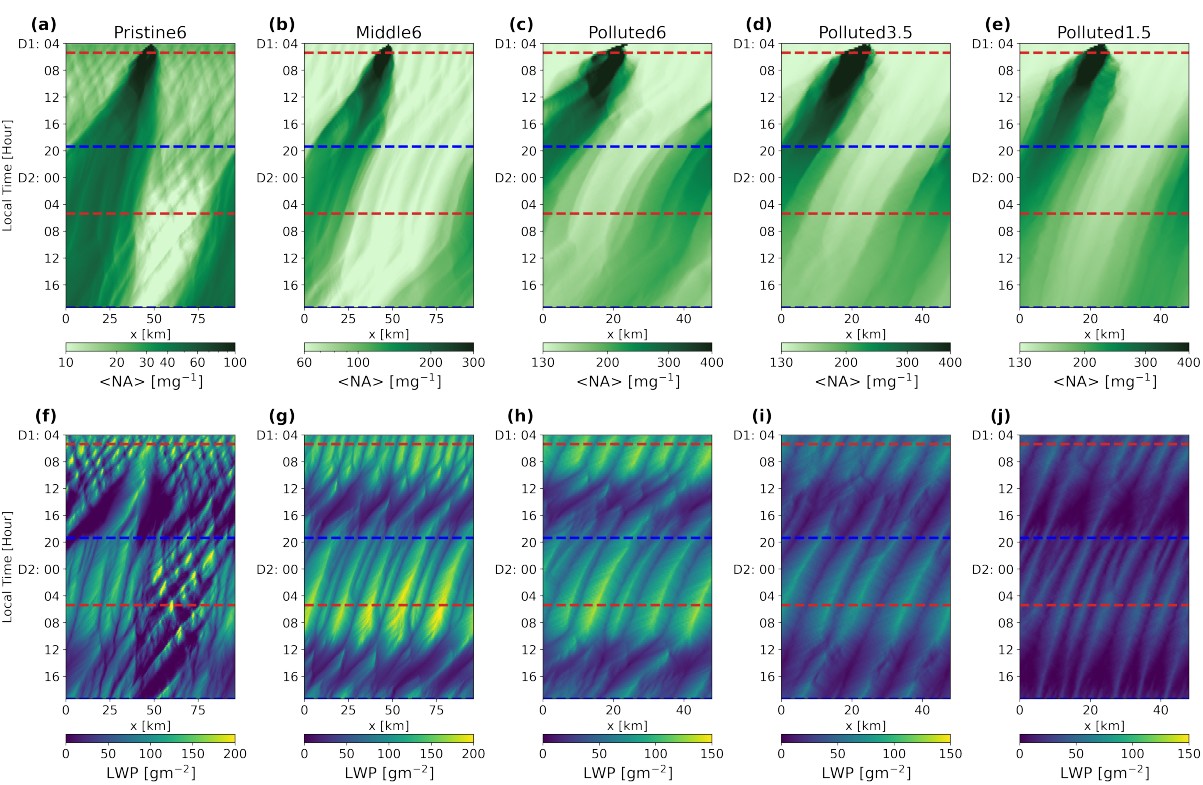

**Figure 5.** Hovmöller plots of (upper row) $\langle N_a \rangle$ and (lower row) LWP in the *Plume* runs for the (first column) Pristine6, (second column) Middle6, (third column) Polluted6, (fourth column) Polluted3.5 and (fifth column) Polluted1.5 cases across the two day simulation. Red and blue dashed lined indicate the times of sunrise and sunset, respectively.





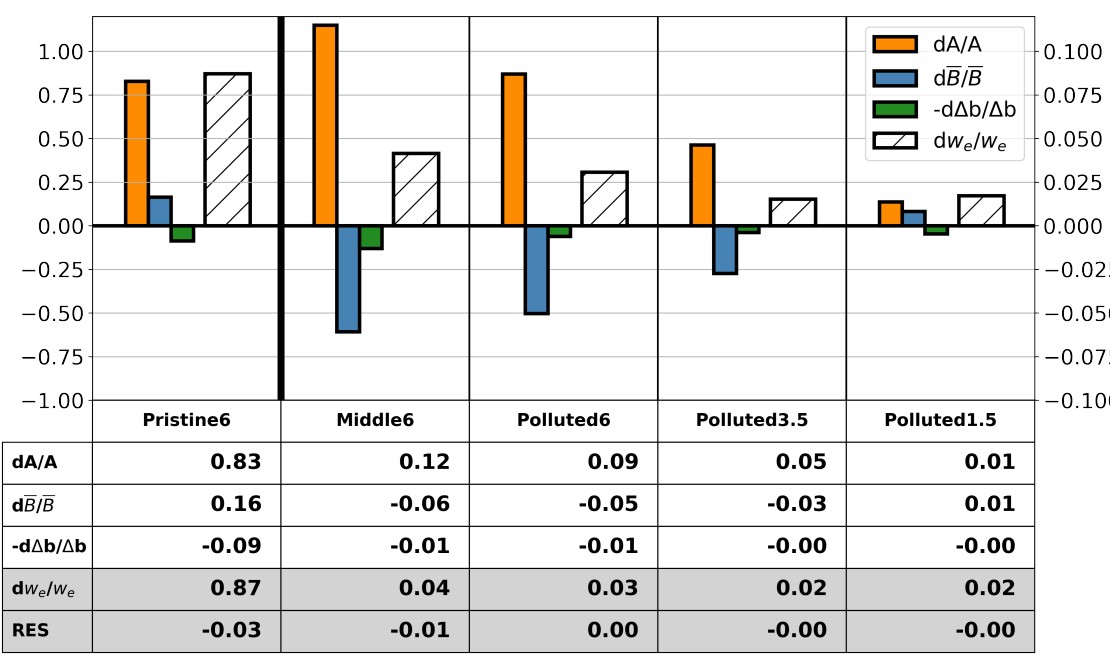

| | Pristine6 | Middle6 | Polluted6 | Polluted3.5 | Polluted1.5 |
|---|---|---|---|---|---|
| dA/A | 0.83 | 0.12 | 0.09 | 0.05 | 0.01 |
| d$\overline{B}$/$\overline{B}$ | 0.16 | -0.06 | -0.05 | -0.03 | 0.01 |
| -dΔb/Δb | -0.09 | -0.01 | -0.01 | -0.00 | -0.00 |
| d$w_e$/$w_e$ | 0.87 | 0.04 | 0.03 | 0.02 | 0.02 |
| RES | -0.03 | -0.01 | 0.00 | -0.00 | -0.00 |

**Figure 6.** Run-averaged fractional changes between the *Plume* and *Ctrl* runs of $dw_e$ (hatched), $dA$ (orange), and $\overline{B}$ (blue) and $\Delta b$ (dark green) in the Pristine6, Middle6, Polluted6, Polluted3.5, and Polluted1.5 cases. Note that the y-axis scale for the Pristine6 case (left side of plot) is ten times greater than that for the other cases (right side of plot). Table below shows the values of the fractional changes and residual between fractional change in $dw_e$ and the sum of the rest of the values.

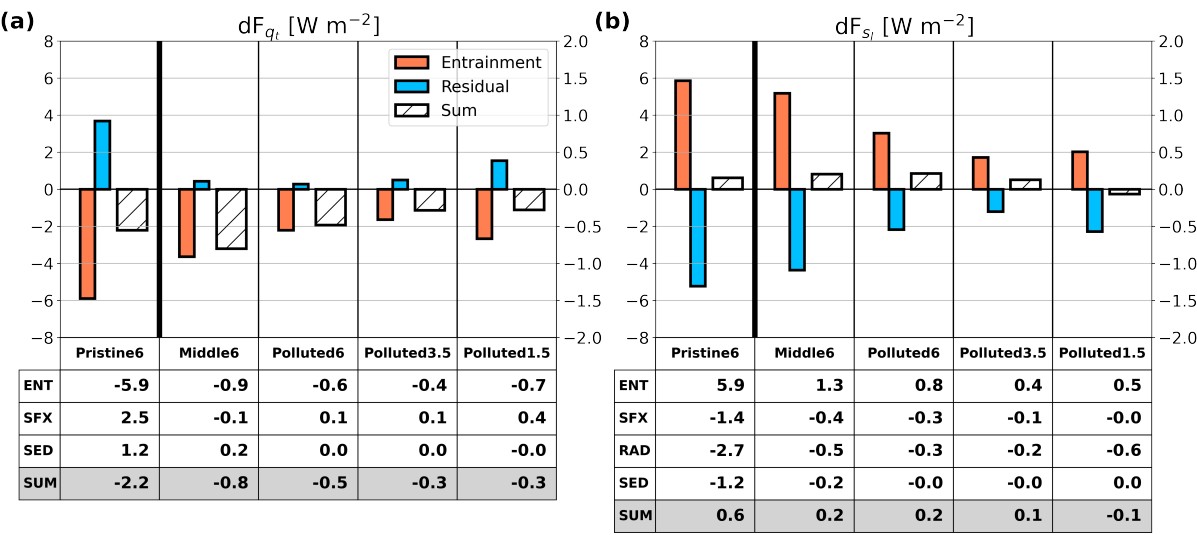

**Figure 7.** (a): Run-averaged differences between the *Plume* and *Ctrl* runs in $q_t$ flux into the boundary layer by entrainment (red) and the sum of the surface flux and surface precipitation (Residual, blue). The bars with hatches represent the sum of all the fluxes. (b): Same as in (a), but for $s_l$. Residuals here include the fluxes induced by radiation responses. The tables below (a) and (b) show the mean values for entrainment (ENT), surface fluxes (SFX), radiation (RAD; for $s_l$ only) and their sum (SUM). Note the logarithmic y-axis scale.

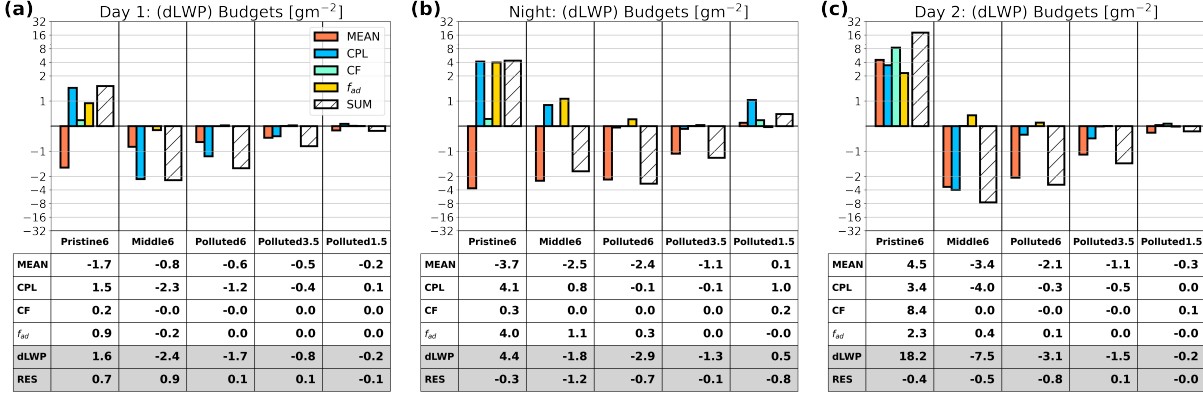

**Figure 8.** $d$LWP budgets for (a) Day 1, (b) Night and (c) Day 2. Bar plots: $d$LWP induced by changes in the MBL mean state (orange) and coupling state (blue) and through changes in CF (green) and $f_{ad}$ (yellow), as well as the sum of all budget terms (SUM, hatched). Tables below show the values given in the plots and residual between the actual $d$LWP and that estimated using an approach shown in Appendix. A. Note the logarithmic y-axis scale.



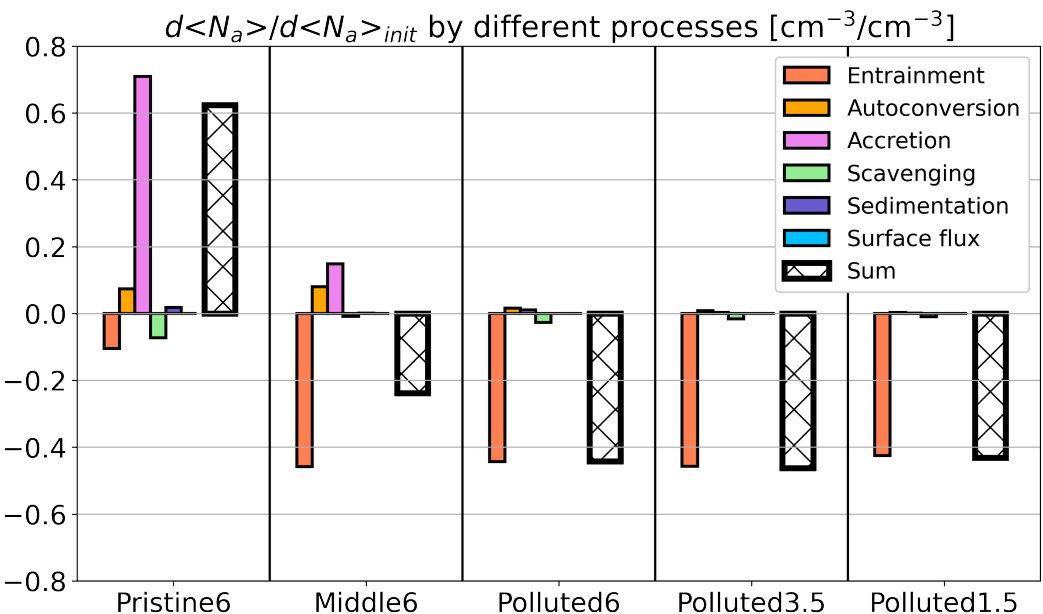

**Figure 9.** Fractional changes in the initial $N_a$ perturbation resulting from different microphysical responses to aerosol injection, averaged across the two day simulation: entrainment (red), autoconversion (orange), accretion (magenta), scavenging (green) and surface flux (blue). Thick bars with hatches represent the sum of all the budget terms.





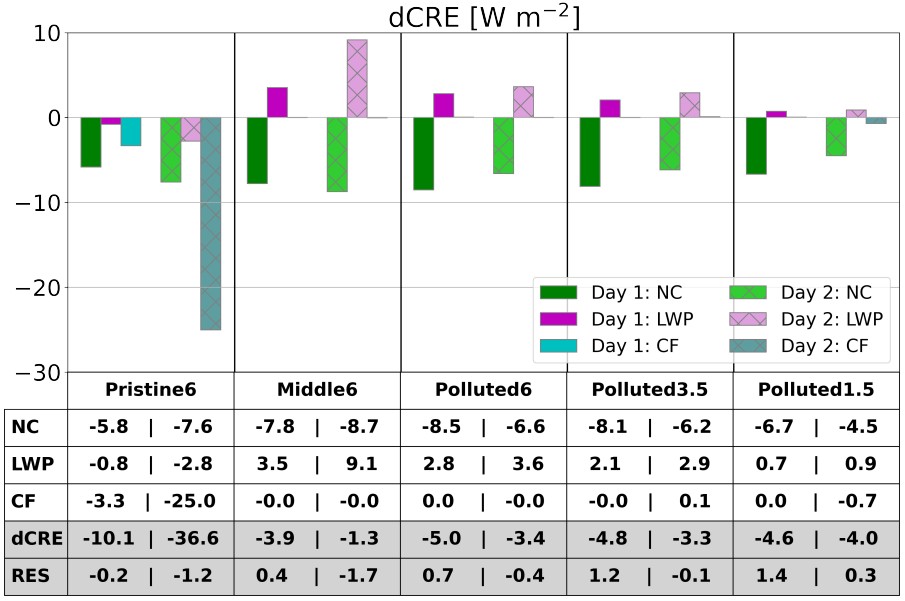

**Figure 10.** The difference in the decomposed 24-hour cloud radiative effect between the *Plume* and *Ctrl* runs ($d$CRE) for each case, showing contributsions from $d$CRE$_{N_c}$ (green), $d$CRE$_{LWP}$ (magenta) and $d$CRE$_{CF}$ (blue). The darker colors show the average $d$CRE for Day 1, and the lighter colors with hatches the averages for Day 2. A table below shows $d$CRE$_{N_c}$, $d$CRE$_{LWP}$, $d$CRE$_{CF}$, $d$CRE, and residual between the actual $d$CRE and that estimated using an approach shown in Appendix B for Day 1 (left) and Day 2 (right).

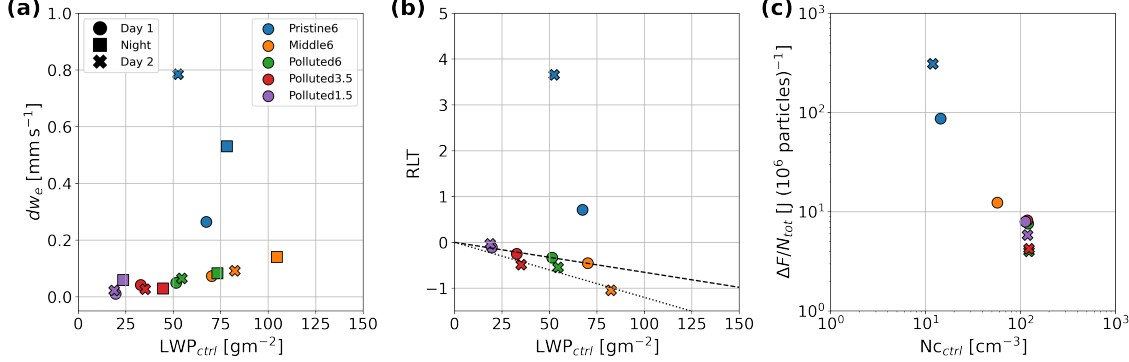

**Figure 11.** (a): Sensitivity of $dw_e$ to LWP$_{ctrl}$. Circle, square and cross markers represent Day 1, Night and Day 2 averages, respectively. (b): Same as (a) but for sensitivity of RLT to LWP$_{ctrl}$. (c): Sensitivity of brightening efficiency (i.e., the radiative forcing per $10^6$ particles) to $N_{c,ctrl}$.





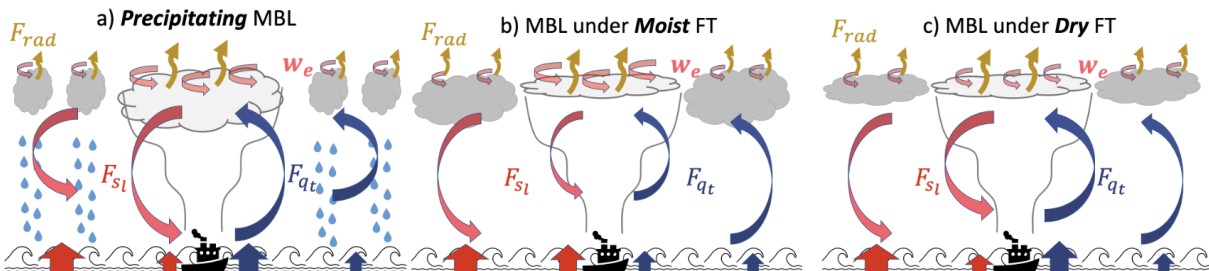

**Figure 12.** This schematic illustrates the responses of entrainment rate (light red arrows), boundary layer turbulence (curved arrows in the boundary layer), surface fluxes (arrows on the surface), and cloud radiative fluxes (yellow arrows on the top of clouds) to aerosol injections. Red arrows represent sensible heat flux, and blue arrows represent moisture flux. The size of the arrows represents the intensity. The center of the domain is the region perturbed by ship tracks; to the left and right of this are the unperturbed regions. Response tendencies are shown for three sets of MBL conditions: (a) precipitating boundary layer, (b) non-precipitating MBL under moist free troposphere, and (c) non-precipitating MBL under dry free troposphere.




**Table 1.** Conditions used for the Pristine6, Middle6, Polluted6, Polluted3.5 and Polluted1.5 cases. The upper panel summarizes the initial and boundary conditions; the middle panel quasi-steady-state conditions in *Ctrl* runs; and the lower panel the information of aerosol injections.

| Case Name | Pristine6 | Middle6 | Polluted6 | Polluted3.5 | Polluted1.5 |
|---|---|---|---|---|---|
| Initial and boundary conditions | | | | | |
| $\langle N_a \rangle$ [cm$^{-3}$] | 20 | 60 | 130 | 130 | 130 |
| $N_{a,FT}$ [cm$^{-3}$] | 50 | 50 | 100 | 100 | 100 |
| $q_{t,FT}$ [g kg$^{-1}$] | 6.0 | 6.0 | 6.0 | 3.5 | 1.5 |
| $D$ [$10^{-6}$ s] | 3.45 | 3.98 | 5.17 | 5.17 | 5.17 |
| Quasi-steady-state conditions in *Ctrl* runs | | | | | |
| $\Delta b$ [m$^2$ s$^{-3}$] | 0.23 | 0.29 | 0.31 | 0.33 | 0.33 |
| $\Delta q_t$ [g kg$^{-1}$] | 3.6 | 3.8 | 3.9 | 5.7 | 7.3 |
| $\Delta s_l$ [K] | 7.7 | 10.3 | 10.7 | 11.4 | 11.4 |
| $NCCLD$ [cm$^{-3}$] | 13 | 56 | 124 | 121 | 117 |
| $\langle N_a \rangle$ [cm$^{-3}$] | 16 | 61 | 133 | 136 | 140 |
| Aerosol injection | | | | | |
| Rate [#/s] | $1 \times 10^{16}$ | $3 \times 10^{16}$ | $3.25 \times 10^{16}$ | $3.25 \times 10^{16}$ | $3.25 \times 10^{16}$ |
| Duration [s] | 914 | 914 | 914 | 914 | 914 |
| Diameter [$\mu$m] | 200 | 200 | 200 | 200 | 200 |
| $d\langle N_a \rangle_{\mathrm{init}}$ [cm$^{-3}$] | 11 | 11 | 74 | 72 | 71 |





**Table 2.** Spatiotemporal averages from the baseline (*Ctrl*) simulations of in-cloud LWP (CLDLWP), cloud fraction (CF), surface precipitation rate ($R_{sfc}$), cloud-base precipitation rate ($R_{cb}$), effective radius of cloud drops ($r_e$), entrainment rate ($w_e$) and inversion height ($z_{inv}$) for the simulated casess. The first, second and third values represent averages across Day 1, Night and Day 2, respectively, with standard deviations shown in square brackets.

| Case | Time | LWPCLD $[\mathrm{g\,m^{-2}}]$ | CF $[\%]$ | $R_{sfc}$ $[\mathrm{mm\,day^{-1}}]$ | $R_{cb}$ $[\mathrm{mm\,day^{-1}}]$ | $r_e$ $[\mu\mathrm{m}]$ | $w_e$ $[\mathrm{mm\,s^{-1}}]$ | $z_{inv}$ $[\mathrm{m}]$ |
|---|---|---|---|---|---|---|---|---|
| Pristine6 | Day 1 | 48.1, [20.3] | 62.5, [15.0] | 0.17, [0.13] | 0.49, [0.31] | 14.52, [0.78] | 0.67, [0.79] | 714.2, [25.8] |
| | Night | 72.0, [13.3] | 87.7, [6.5] | 0.30, [0.14] | 0.83, [0.23] | 15.49, [0.29] | 2.66, [0.56] | 697.6, [23.0] |
| | Day 2 | 30.2, [7.8] | 52.5, [11.1] | 0.14, [0.14] | 0.37, [0.26] | 14.28, [0.80] | 0.95, [0.84] | 652.2, [21.8] |
| Middle6 | Day 1 | 75.3, [30.0] | 99.4, [0.6] | 0.01, [0.01] | 0.07, [0.05] | 10.74, [0.71] | 2.54, [0.79] | 752.4, [11.9] |
| | Night | 105.6, [15.5] | 99.9, [0.1] | 0.01, [0.01] | 0.11, [0.03] | 11.36, [0.30] | 4.01, [0.11] | 757.7, [8.5] |
| | Day 2 | 82.1, [38.1] | 99.2, [0.7] | 0.02, [0.02] | 0.11, [0.09] | 11.12, [0.77] | 2.44, [0.82] | 761.6, [13.0] |
| Polluted6 | Day 1 | 56.0, [24.7] | 99.4, [0.6] | 0.00, [<0.01] | 0.01, [0.01] | 8.13, [0.58] | 2.31, [0.67] | 705.4, [19.1] |
| | Night | 76.1, [11.9] | 99.9, [0.0] | 0.00, [<0.01] | 0.02, [<0.01] | 8.63, [0.23] | 3.70, [0.09] | 691.2, [17.5] |
| | Day 2 | 54.9, [25.1] | 99.3, [0.9] | 0.00, [<0.01] | 0.01, [0.01] | 8.02, [0.60] | 2.51, [0.58] | 672.2, [15.9] |
| Polluted3.5 | Day 1 | 34.9, [11.9] | 96.6, [2.7] | -0.00, [<0.01] | 0.01, [<0.01] | 7.51, [0.43] | 2.87, [0.57] | 748.2, [14.8] |
| | Night | 45.7, [6.9] | 98.7, [0.8] | -0.00, [<0.01] | 0.01, [<0.01] | 7.89, [0.22] | 4.16, [0.18] | 735.4, [12.8] |
| | Day 2 | 35.1, [12.8] | 96.2, [3.1] | -0.00, [<0.01] | 0.01, [<0.01] | 7.40, [0.45] | 2.95, [0.60] | 728.9, [13.8] |
| Polluted1.5 | Day 1 | 18.7, [9.2] | 74.8, [18.3] | -0.00, [<0.01] | 0.01, [<0.01] | 6.74, [0.51] | 2.33, [0.82] | 754.2, [20.2] |
| | Night | 23.1, [5.2] | 84.2, [6.3] | -0.00, [<0.01] | 0.01, [<0.01] | 7.02, [0.27] | 4.04, [0.09] | 731.9, [18.8] |
| | Day 2 | 15.8, [8.7] | 67.7, [21.4] | -0.00, [<0.01] | 0.00, [<0.01] | 6.46, [0.50] | 2.35, [0.96] | 717.2, [20.2] |





**Table 3.** As in Table 2, but for the differences between the *Plume* and *Ctrl* runs averaged across (first) Day 1, (second) Night and (third) Day 2.

| Case | Time | $d$LWPCLD [$\mathrm{g\,m^{-2}}$] | $d$CF [%] | $dR_{sfc}$ [$\mathrm{mm\,day^{-1}}$] | $dR_{cb}$ [$\mathrm{mm\,day^{-1}}$] | $dr_e$ [$\mu$m] | $dw_e$ [$\mathrm{mm\,s^{-1}}$] | $dz_{inv}$ [m] |
|---|---|---|---|---|---|---|---|---|
| Pristine6 | Day 1 | 1.8, [3.9] | 1.9, [7.0] | -0.02, [ 0.03] | -0.06, [ 0.05] | -1.26, [0.29] | 0.46, [0.19] | 3.8, [4.3] |
| | Night | 4.0, [8.5] | 2.9, [6.5] | -0.09, [ 0.05] | -0.26, [ 0.08] | -1.72, [0.15] | 0.72, [0.46] | 12.5, [5.4] |
| | Day 2 | 18.1, [7.8] | 25.4, [7.4] | -0.01, [ 0.05] | -0.04, [ 0.10] | -1.56, [0.23] | 0.85, [0.24] | 50.3, [9.0] |
| Middle6 | Day 1 | -2.6, [1.8] | 0.0, [0.1] | 0.00, [<0.01] | -0.01, [<0.01] | -0.86, [0.11] | 0.11, [0.09] | 2.4, [0.8] |
| | Night | -1.8, [0.8] | 0.0, [0.0] | 0.00, [<0.01] | -0.03, [ 0.01] | -0.93, [0.04] | 0.16, [0.12] | 4.6, [0.9] |
| | Day 2 | -7.6, [1.9] | 0.0, [0.2] | 0.01, [<0.01] | -0.04, [ 0.03] | -1.26, [0.11] | 0.08, [0.20] | 12.5, [1.4] |
| Polluted6 | Day 1 | -1.8, [0.7] | -0.1, [0.2] | 0.00, [<0.01] | 0.00, [<0.01] | -0.68, [0.04] | 0.09, [0.04] | 1.5, [0.4] |
| | Night | -2.8, [1.0] | 0.0, [0.0] | 0.00, [<0.01] | 0.00, [<0.01] | -0.75, [0.02] | 0.12, [0.02] | 3.5, [0.7] |
| | Day 2 | -3.0, [2.0] | 0.1, [0.2] | 0.00, [<0.01] | 0.00, [<0.01] | -0.64, [0.08] | 0.08, [0.10] | 5.5, [0.6] |
| Polluted3.5 | Day 1 | -0.9, [0.6] | -0.0, [0.2] | 0.00, [<0.01] | 0.00, [<0.01] | -0.55, [0.03] | 0.06, [0.02] | 1.2, [0.4] |
| | Night | -1.2, [0.3] | 0.0, [0.1] | 0.00, [<0.01] | 0.00, [<0.01] | -0.63, [0.02] | 0.05, [0.02] | 2.1, [0.2] |
| | Day 2 | -1.5, [0.6] | -0.2, [0.3] | 0.00, [<0.01] | 0.00, [<0.01] | -0.55, [0.06] | 0.03, [0.03] | 3.0, [0.2] |
| Polluted1.5 | Day 1 | -0.2, [0.3] | 0.0, [1.2] | 0.00, [<0.01] | 0.00, [<0.01] | -0.50, [0.03] | 0.03, [0.05] | 0.9, [0.3] |
| | Night | 0.5, [0.3] | 1.9, [0.5] | 0.00, [<0.01] | 0.00, [<0.01] | -0.51, [0.02] | 0.08, [0.03] | 1.5, [0.4] |
| | Day 2 | -0.2, [0.4] | 0.7, [0.9] | 0.00, [<0.01] | 0.00, [<0.01] | -0.43, [0.07] | 0.03, [0.06] | 3.7, [0.3] |