# Peer review of "Microphysical, macrophysical and radiative responses of subtropical marine clouds to aerosol injections."

_Atmospheric Chemistry and Physics, 2022_

## Referee Comment (RC2)

**Comments to the manuscript with ID "acp-2022-351"**

**General comments:**

This study explored the stratocumulus marine boundary-layer (MBL) and cloud responses to aerosol injections using idealized large-eddy simulations to address the longstanding aerosolcloud interactions. It shows that "the cloud-top entrainment rate, MBL turbulence, surface fluxes and cloud microphysics differently depending on meteorological conditions." These findings can contribute to the effort of disentangling aerosol effects on cloud properties from the meteorological effects. In addition, this study developed methods to quantitatively decompose LWP adjustments into contributions by different processes, and the cloud radiative effect to contributions from Nc, LWP, and cloud fraction.

Overall, the manuscript is well written and organized with clear scientific questions being challenged and answered. Therefore, I recommend the publication of this manuscript with minor revision.

**Specific comments:**

L.5-10: By just reading this sentence, "The key results are that (a) the cloud top entrainment rate increases in all cases, with stronger increases for thicker than thinner clouds", it is unclear to me what causes "cloud top entrainment rate increases". Since this is in the abstract, I suggest the author elaborate on the causality to improve the presentation and readability.

L.111-119: What are the parameters for the input aerosol size distribution? Are they from insitu measurement or arbitrary?

L.125-130: Are the large-scale forcings from ERA5 reanalysis? How did you validate them? By saying "steady forcings", did you mean the large-scale advective tendencies of thermodynamics and divergence do not vary with time? If so, could this affect the boundary structures and therefore the cloud macrophysics?

L.132-136: Are the aerosols uniformly distributed in the entire domain in the control simulation? Are the injected aerosols instantaneously distributed into the domain after the injection?

L.138-139: Do you apply nudging to large-scale divergence? If so, what is the reference divergence? Does Na, FT varies with time? Why do you need to adjust it? Would such adjusting affect your conclusion on aerosol-cloud interactions?

L.144-149: Why not use the same domain size for all simulations (96\*9.6 km)? The convective organization could be different between the 96\*9.6 km and 48\*9.6 km-sized domain, which may result in different LWP and CF. Could the authors check this at least for one set of simulations to make exclude the effect of domain size difference?

L.537-539: Since the cloud responses to aerosols strongly depends on meteorology conditions, would it be necessary to validate the meteorological conditions in the presented simulations using observational or reanalysis data? Such a validation could strengthen the conclusion.

L.435: Can the authors elaborate more on the "coupling and surface flux responses"?

Technical comments:

L.64: Is the deviation missing after the "-3.7 W/m^2"?

L.193: Is "s\_l" defined?

Fig.3 caption: Since the authors labeled the each subplot, why not use labels to improve the readability, e.g.,  $q_t$ : (a) and (c), ... ?

L.213: Does "run-average" mean time average after the spin-up time?

L.569: Can the authors rephrase the sentence? I am confused with the parenthesis and "runs with runs".

---

## Author Comment (AC1)

Referee #1

Major Comments

1. **Comment**: *The three polluted cases use a smaller domain than the "pristine" and "middle" cases. The authors explain in lines 146-149 that the wider domain for the pristine and middle case is for mesoscale circulation, which is more significant in the precipitation cases. By comparing the roll cloud size between the middle and polluted case, it looks similar if the "mesoscale circulation" refers to the mesoscale circulation which maintains the formation of roll structure. So, I am not quite convinced that a different domain size is necessary. I recommend using 96 km X 9.6 km for all the simulations so that many analyses using domain and run averaged values can be more robust.*

**Reply**: The authors agree that it would be ideal for all runs to have the same domain size. However, due to limited computational cost, we limited the use of the larger domain to the cleaner cases, as these were the case with some precipitation, where capturing mesoscale circulations might be important. Previous studies have shown that strong precipitation gradients in response to aerosol injection can induce change in the mesoscale circulation (e.g., Wang et al 2011). In the plume run for the Pristine6 case, the plume region has overcast clouds, but there is no cloud at the edges of the plume (Figs.5a,f) as a result of the induced mesoscale circulation. In the Middle and Polluted cases, on the other hand, there is no notable difference between the plume and background region (Figs.5b-e and g-j).

The reason for the wider domain for the Pristine case is to consider the impact of the induced mesoscale circulation due to aerosol injection. If the domain were as small as the Polluted cases, the plume would be overcast by the end of simulation and thus, the induced mesoscale circulation would change. Since there is no (or a very little) 'change' in mesoscale circulation in the Polluted cases, we assume that the width of the domain does not have a significant impact on the result.

2. **Comment**: *Section 3.2.4 shows the budget analysis of the impacts of 5 different processes on the cloud number concentration. But the method to decompose the droplet number concentration is omitted. Given the importance of the information, I recommend including the method in detail either in the main context or in the appendix.*

**Reply**: The details of the methods to derive the budget for the aerosol number concentration (not cloud number concentration) are shown in Line 111-119 in the revised manuscript (hereafter, "Line" in our Reply indicates line number in the revised manuscript, unless it is specified). One process that is not illustrated here is entrainment. We have added this to the revised manuscript (Line 385-387)

3. **Comment**: Line 215-216: *"the low A may be attributed to the low qc,inv caused by the high sedimentation velocity of large cloud droplets...."* How about the role of cloud thickness in low A?

**Reply**: We agree that it seems reasonable to expect that reduced cloud thickness might drive a decrease in A but the data do not indicate this. The cloud thickness in the Pristine6 case is greater than in the Polluted3.5 and 1.5 cases (Table 2), but A in the Pristine6 case is much lower (Fig.4). This indicates that cloud water content at cloud top, immediately adjacent to the entrainment zone below the free troposphere, is more sensitive to the droplet sedimentation velocity than to cloud thickness.

Minor Comment

1. **Line 109: Above? Below?**

Ice phase hydrometeor species are not required because the simulation domain is everywhere above the freezing temperature.

2. **Line 138-139: How do you adjust the free-tropospheric aerosol and divergence?**

We multiply the initial free-tropospheric aerosol number concentration and divergence by spatially-uniform factors.

3. **Line 152: any reference for the number 10.5 m/s?**

We added a sentence to support that the wind speed is reasonable "This wind speed is similar to the near-surface wind speed in Blossey et al. (2013) (8.3 m/s), and allows for a moderate increase in winds with altitude in the MBL." (Line 156-157)

4. **Line 177: "this reduces the primary source of turbulence in the marine low clouds (Table 2)". Which variable from table 2 do you use to analyze turbulence? I don't find TKE or other variables that represent turbulence.**

We edited the sentence in the revised manuscript. "Table2" is in reference to the reduced LWP, not the TKE.

5. **Equation 1 please include the mathematical equation for entrainment efficiency A to show how A relates to include liquid water amount:**

We calculate entrainment efficiency (**A**) using eq.(3) in the revised manuscript (i.e., just an arrangement of eq.2). This is why Line (215-220) is added to explain that our calculation of **A** could result in errors embedded in equation 3. We have added "As approximated by equations (5) and (6) in Bretherton et al (2007)," (Line 213) in order to explain how **A** can be mathematically approximated based on microphysics.

6. ***Line 267-270: reduction in re leads to the changes in dA/A for all the cases. Are the processes associated with this relationship the same for all the cases?***

The same processes are at play in all cases, but the relative importance of each process in driving the changes in entrainment efficiency differs from case to case, as described in the text following lines 267-270. As such, no change has been made to the text.

7. ***Line 280-281: explain why stronger entrainment tends to sharpen the inversion. Entrainment leads to the mixing between the air above and below invesion, which is supposed to smooth the boundary at the inversion level.***

This is because, to first order (i.e. assuming the boundary layer is well-mixed and the free tropospheric temperature profile is a fixed function of height) a deeper MBL will have a colder temperature at the base of the inversion. Thus the temperature jump across the inversion is expected to increase as the inversion height increases, in a well-mixed MBL.

8. ***Line 303-305: I assume this is referred from Figure 8, which has not been explained yet. Add "Figure 8" to help the readers to digest.***

It refers to the values in the table below, which has been explained earlier.

9. ***Is it better to exchange the order of section 3.2.3 and section 3.2.2 for the purpose of organizing the paper? The contribution from decoupling is introduced in 3.2.3 but already discussed in 3.2.2.***

I would think that section 3.2.2 might be better to be put earlier than 3.2.3, because perturbation in qt and sl in the cloud layer determines zcb, which affects the liquid water path adjustment. Actually, "more significant turbulence/more stratification" are first introduced in Section 3.2.1 (i.e., change in $\bar{B}$ ). To treat the issue that the coupling/decoupling is introduced in 3.2.3, but discussed in 3.2.2, we change the heading of section 3.2.2 to "Perturbations in qt and sl fluxes into the MBL by aerosol injections" and this section mainly illustrates net fluxes of qt and sl into the boundary layer. Coupling/decoupling is introduced/mainly discussed in 3.2.3.

10. ***Line 594: clarify how to calculate fad the adiabaticity.***

It is provided in the next sentence (Line 596-597).

11. ***Line 650: cloud optical thickness? Albedo?***

Thank you for noting this. The words "cloud optical thickness" have been changed to "cloud albedo".

Referee #2

*L. 5-10: By just reading this sentence, "The key results are that (a) the cloud top entrainment rate increases in all cases, with stronger increases for thicker than thinner clouds", it is unclear to me what causes "cloud top entrainment rate increases". Since this is in the abstract, I suggest the author elaborate on the causality to improve the presentation and readability.*

The sentence is rephrased as "the cloud top entrainment increases with aerosol injections in all cases due to enhanced entrainment efficiency, with stronger entrainment increases for thicker than thinner clouds".

*L.111-119: What are the parameters for the input aerosol size distribution? Are they from in- situ measurement or arbitrary?*

The parameters for the input aerosol size distribution are chosen to be similar to sea-salt spray and background aerosol properties (see Fig.2 and text related to it).

*L.125-130: Are the large-scale forcings from ERA5 reanalysis?*

The forcings are from an idealized case whose soundings are based on ERA Interim reanalysis and whose large-scale forcings are more idealized (Zhang et al, 2012, sec 2 and 3.1-3.2) with modifications for the LES setup in Blossey et al (2013, appendix A4-A5).

*How did you validate them?*

While the process of validating an idealized, climatological case study is not so clear, the properties of clouds and the boundary layer (cloud thickness and amount, boundary layer depth, etc) simulated here lie within the range of those observed for this location during summertime.

We would also note that this idealized case study of well-mixed coastal stratocumulus (CGILS S12) is established in the literature, having been used to study low cloud feedbacks (Blossey et al, 2013; Bretherton et al, 2013; Zhang et al, 2013; Blossey et al, 2016), mesoscale cloud organization (Kazil et al, 2017), the internal response time scales of a stratocumulus-capped boundary layer (Jones et al, 2014) and the diurnal breakup of cloud over land (Ghonima et al, 2016).

The ability to include a diurnal cycle of radiation and to directly compute shortwave cloud radiative effects makes this case preferable to the more simple DYCOMS RF01 or RF02 (Stevens et al, 2005; Ackerman et al, 2009) cases that have been used elsewhere in the aerosol-cloud interaction literature.

*By saying "steady forcings", did you mean the large-scale advective tendencies of thermodynamics and divergence do not vary with time?*

Yes, the large-scale advective tendences and subsidence do not vary with time.

***If so, could this affect the boundary structures and therefore the cloud macrophysics?***

As noted in the text, "such an air mass would be expected to experience changing forcings over that time period", and we expect that time-varying forcings (e.g., stratocumulus-to-cumulus transition due to change in SST, large-scale divergence…) would affect cloud macrophysics in the real atmosphere. This is acknowledged by noting that "In future work, we plan to evaluate the effect of aerosol injection in cases with evolving large-scale conditions, e.g., in Lagrangian case studies of stratocumulus to cumulus transitions (Sandu and Stevens, 2011; Blossey et al., 2021)." The reason for using fixed forcings is given in the text, i.e. that "…we choose steady forcings to characterize the effect of aerosol injection over time on an important MBL cloud regime." Adding a diurnal cycle itself already introduces some quite complicated unsteadiness, and we felt that adding time-varying forcings for this first study of aerosol injection from our group would make it more straightforward to identify the influences of different meteorological drivers.

For even better clarity, we have edited this text to now read: "… we choose steady forcings in order to be able to both characterize and more clearly attribute to key processes the effect of aerosol injection in the presence of a diurnal cycle on an important MBL cloud regime."

***L.132-136: Are the aerosols uniformly distributed in the entire domain in the control simulation?***

The initial aerosol concentration is horizontally-uniform, with one value in the boundary layer $<Na>$ and a second in the free troposphere $Na,FT$, as specified in Table 1. While some variability of the aerosol develops during the 12 hour spin up period before aerosol injection, the boundary-layer-mean $Na$ is nearly steady during this time (Fig.S1g in Supplementary Material).

***Are the injected aerosols instantaneously distributed into the domain after the injection?***

The aerosol injection is imposed as a localized perturbation of the aerosol surface flux which is localized in space. This will place the aerosols in the bottom-most grid level immediately after injection. Subsequently, the injected aerosols are transported and mixed by the resolved-scale flow field and the model's subgrid-scale mixing parameterization.

As can be seen in Figure 5, there is some variation in the background aerosol concentration due to interactions with the unperturbed clouds. As also seen in Figure 5, for the injection cases a single sprayer moves in the y direction one time through the middle of the x axis. The sprayer travels only for the first 15 minutes after spin up. The injected aerosols spread laterally with time, as shown in Fig.5

***L.138-139: Do you apply nudging to large-scale divergence? If so, what is the reference divergence?***

No. The large-scale divergence/subsidence is constant with time in each simulation.

***Does Na, FT vary with time? Why do you need to adjust it?***

We use a slightly different q_t, NA_ft, and large-scale divergence for each case (Table 1), but they do not vary with time.

***Would such adjusting affect your conclusion on aerosol-cloud interactions?***

Without the adjustment to NA_ft, the aerosol concentration would be changing with time in the control simulation. This would complicate the interpretation of cloud changes in response to the aerosol perturbation.

***L.537-539: Since the cloud responses to aerosols strongly depends on meteorology conditions, would it be necessary to validate the meteorological conditions in the presented simulations using observational or reanalysis data? Such a validation could strengthen the conclusion.***

We have added some texts to validate our results based on observations. (e.g, Line 464-469, 560-561)

***L.435: Can the authors elaborate more on the "coupling and surface flux responses"?***

The sentence has been edited to now read:

"This is due to the greater entrainment enhancement under a moist FT, leading to more significant stratification and a weaker surface flux response than under a dry FT, as described above in earlier sections."

***L.64: Is the deviation missing after the "-3.7 W/m^2"?***

Thank you for pointing it out. We have changed it to "exert a cloud radiative effect perturbation of -3.7 W/m2"

***L.193: Is "s_l" defined?***

Thank you for noticing this. The definition was omitted and has now been added (Line 101).

***Fig.3 caption: Since the authors labeled each subplot, why not use labels to improve the readability, e.g., q_t: (a) and (c), ... ?***

We have changed the figure as suggested. Thank you for your suggestion.

***L.213: Does "run-average" mean time average after the spin-up time?***

Yes. The sentence has been edited to clarify this. (Line 215)

***L.569: Can the authors rephrase the sentence? I am confused with the parenthesis and "runs with runs".***

Thank you for noticing this error. The wording has been edited.

- M. Zhang, C. S. Bretherton, P. N. Blossey, Sandrine Bony, Florent Brient and Jean-Christophe Golaz, 2012. The CGILS Experimental Design to Investigate Low Cloud Feedbacks in General Circulation Models by Using Single-Column and Large-Eddy Simulation Models. J. Adv. Model. Earth Syst., Vol. 4, M12001, doi:10.1029/2012MS000182.
- P. N. Blossey, C. S. Bretherton, M. Zhang, A. Cheng, S. Endo, T. Heus, Y. Liu, A. Lock, S. R. de Roode and K.-M. Xu, 2013. Marine low cloud sensitivity to an idealized climate change: The CGILS LES Intercomparison. J. Adv. Model. Earth Syst., 5, 234-258, doi:10.1002/jame.20025.
- C. S. Bretherton, P. N. Blossey and C. R. Jones, 2013. Mechanisms of marine low cloud sensitivity to idealized climate perturbations: A single-LES exploration extending the CGILS cases. J. Adv. Model. Earth Syst., 5, 316-337, doi:10.1002/jame.20019.
- M. Zhang and co-authors, 2013. CGILS: Results from the first phase of an international project to understand the physical mechanisms of low cloud feedbacks in general circulation models. J. Adv. Model. Earth Sys., 5, 826-842, doi:10.1002/2013MS000246.
- C. R. Jones, C. S. Bretherton and P. N. Blossey, 2014. Fast Stratocumulus Timescale in Mixed Layer Model and Large Eddy Simulation. J. Adv. Model. Earth Syst., 6, 206-222, doi:10.1002/2013MS000289
- P. N. Blossey, C. S. Bretherton, A. Cheng, S. Endo, T. Heus, A. Lock and J. J. van der Dussen, 2016. CGILS Phase 2 LES intercomparison of response of subtropical marine low cloud regimes to CO2 quadrupling and a CMIP3-composite forcing change. J. Adv. Model. Earth Syst., 08, doi:10.1002/2016MS000765
- Ghonima, M. S., Heus, T., Norris, J. R., & Kleissl, J. (2016). Factors Controlling Stratocumulus Cloud Lifetime over Coastal Land, Journal of the Atmospheric Sciences, 73(8), 2961-2983.
- Kazil, J., Yamaguchi, T., and Feingold, G. (2017), Mesoscale organization, entrainment, and the properties of a closed-cell stratocumulus cloud, J. Adv. Model. Earth Syst., 9, 2214– 2229, doi:10.1002/2017MS001072.